# Online Reinforcement Learning in Stochastic Games

**Chen-Yu Wei**
Institute of Information Science
Academia Sinica, Taiwan
`bahh723@iis.sinica.edu.tw`

**Yi-Te Hong**
Institute of Information Science
Academia Sinica, Taiwan
`ted0504@iis.sinica.edu.tw`

**Chi-Jen Lu**
Institute of Information Science
Academia Sinica, Taiwan
`cjlu@iis.sinica.edu.tw`

## Abstract

We study online reinforcement learning in average-reward stochastic games (SGs). An SG models a two-player zero-sum game in a Markov environment, where state transitions and one-step payoffs are determined simultaneously by a learner and an adversary. We propose the UCSG algorithm that achieves a sublinear regret compared to the game value when competing with an arbitrary opponent. This result improves previous ones under the same setting. The regret bound has a dependency on the *diameter*, which is an intrinsic value related to the mixing property of SGs. If we let the opponent play an optimistic best response to the learner, UCSG finds an $\varepsilon$-maximin stationary policy with a sample complexity of $\tilde{\mathcal{O}}\left(\text{poly}(1/\varepsilon)\right)$, where $\varepsilon$ is the gap to the best policy.

## 1 Introduction

Many real-world scenarios (e.g., markets, computer networks, board games) can be cast as multi-agent systems. The framework of Multi-Agent Reinforcement Learning (MARL) targets at learning to act in such systems. While in traditional reinforcement learning (RL) problems, Markov decision processes (MDPs) are widely used to model a single agent's interaction with the environment, stochastic games (SGs, [32]), as an extension of MDPs, are able to describe multiple agents' simultaneous interaction with the environment. In this view, SGs are most well-suited to model MARL problems [24].

In this paper, two-player zero-sum SGs are considered. These games proceed like MDPs, with the exception that in each state, both players select their own actions *simultaneously* [1], which jointly determine the transition probabilities and their rewards . The *zero-sum* property restricts that the two players' payoffs sum to zero. Thus, while one player (Player 1) wants to maximize his/her total reward, the other (Player 2) would like to minimize that amount. Similar to the case of MDPs, the reward can be discounted or undiscounted, and the game can be episodic or non-episodic.

In the literature, SGs are typically learned under two different settings, and we will call them *online* and *offline* settings, respectively. In the offline setting, the learner controls both players in a centralized manner, and the goal is to find the equilibrium of the game [33, 21, 30]. This is also known as finding the worst-case optimality for each player (a.k.a. maximin or minimax policy). In this case, we care about the *sample complexity*, i.e., how many samples are required to estimate the worst-case optimality such that the error is below some threshold. In the online setting, the learner controls only one of the players, and plays against an arbitrary opponent [24, 4, 5, 8, 31]. In this case, we care

about the learner's *regret*, i.e., the difference between some benchmark measure and the learner's total reward earned in the learning process. This benchmark can be defined as the total reward when both players play optimal policies [5], or when Player 1 plays the best stationary response to Player 2 [4]. Some of the above online-setting algorithms can find the equilibrium simply through self-playing.

Most previous results on offline sample complexity consider discounted SGs. Their bounds depend heavily on the chosen discount factor [33, 21, 30, 31]. However, as noted in [5, 19], the discounted setting might not be suitable for SGs that require long-term planning, because only finite steps are relevant in the reward function it defines. This paper, to the best of our knowledge, is the first to give an offline sample complexity bound of order $\tilde{\mathcal{O}}\left(\text{poly}(1/\varepsilon)\right)$ in the average-reward (undiscounted and non-episodic) setting, where $\varepsilon$ is the error parameter. A major difference between our algorithm and previous ones is that the two players play asymmetric roles in our algorithm: by focusing on finding only one player's worst-case optimal policy at a time, the sampling can be rather efficient. This resembles but strictly extends [13]'s methods in finding the maximin action in a two-stage game.

In the online setting, we are only aware of [5]'s R-MAX algorithm that deals with average-reward SGs and provides a regret bound. Considering a similar scenario and adopting the same regret definition, we significantly improve their bounds (see Appendix A for details). Another difference between our algorithm and theirs is that ours is able to output a currently best stationary policy at any stage in the learning process, while theirs only produces a $T_\varepsilon$-step fixed-horizon policy for some input parameter $T_\varepsilon$. The former could be more natural since the worst-case optimal policy is itself a stationary policy.

The techniques used in this paper are most related to RL for MDPs based on the optimism principle [2, 19, 9] (see Appendix A). The optimism principle built on concentration inequalities automatically strikes a balance between exploitation and exploration, eliminating the need to manually adjust the learning rate or the exploration ratio. However, when importing analysis from MDPs to SGs, we face the challenge caused by the opponent's uncontrollability and non-stationarity. This prevents the learner from freely exploring the state space and makes previous analysis that relies on stationary distribution's perturbation analysis [2] useless. In this paper, we develop a novel way to replace the opponent's non-stationary policy with a stationary one in the analysis (introduced in Section 5.1), which facilitates the use of techniques based on perturbation analysis. We hope that this technique can benefit future analysis concerning non-stationary agents in MARL.

One related topic is the robust MDP problem [29, 17, 23]. It is an MDP where some state-action pairs have adversarial rewards and transitions. It is often assumed in robust MDP that the adversarial choices by the environment are not directly observable by the Player, but in our SG setting, we assume that the actions of Player 2 can be observed. However, there are still difficulties in SG that are not addressed by previous works on robust MDP.

Here we compare our work to [23], a recent work on learning robust MDP. In their setting, there are adversarial and stochastic state-action pairs, and their proposed OLRM2 algorithm tries to distinguish them. Under the scenario where the environment is fully adversarial, which is the counterpart to our setting, the worst-case transitions and rewards are all revealed to the learner, and what the learner needs to do is to perform a maximin planning. In our case, however, the worst-case transitions and rewards are still to be learned, and the opponent's arbitrary actions may hinder the learner to learn this information. We would say that the contribution of [23] is orthogonal to ours.

Other lines of research that are related to SGs are on MDPs with adversarially changing reward functions [11, 27, 28, 10] and with adversarially changing transition probabilities [35, 1]. The assumptions in these works have several differences with ours, and therefore their results are not comparable to our results. However, they indeed provide other viewpoints about learning in stochastic games.

## 2 Preliminaries

**Game Models and Policies.** A SG is a 4-tuple $M = (\mathcal{S}, \mathcal{A}, r, p)$. $\mathcal{S}$ denotes the state space and $\mathcal{A} = \mathcal{A}^1 \times \mathcal{A}^2$ the players' joint action space. We denote $S = |\mathcal{S}|$ and $A = |\mathcal{A}|$. The game starts from an initial state $s_1$. Suppose at time $t$ the players are at state $s_t$. After the players play the joint actions $(a_t^1, a_t^2)$, Player 1 receives the reward $r_t = r(s_t, a_t^1, a_t^2) \in [0, 1]$ from Player 2, and both players visit state $s_{t+1}$ following the transition probability $p(\cdot|s_t, a_t^1, a_t^2)$. For simplicity, we consider

deterministic rewards as in [3]. The extension to stochastic case is straightforward. We shorten our notation by $a := (a^1, a^2)$ or $a_t := (a_t^1, a_t^2)$, and use abbreviations such as $r(s_t, a_t)$ and $p(\cdot|s_t, a_t)$.

Without loss of generality, players are assumed to determine their actions based on the history. A policy $\pi$ at time $t$ maps the history up to time $t$, $H_t = (s_1, a_1, r_1, ..., s_t) \in \mathcal{H}_t$, to a probability distribution over actions. Such policies are called *history-dependent* policies, whose class is denoted by $\Pi^{\mathrm{HR}}$. On the other hand, a *stationary* policy, whose class is denoted by $\Pi^{\mathrm{SR}}$, selects actions as a function of the current state. For either class, joint policies $(\pi^1, \pi^2)$ are often written as $\pi$.

**Average Return and the Game Value.** Let the players play joint policy $\pi$. Define the $T$-step total reward as $R_T(M, \pi, s) := \sum_{t=1}^{T} r(s_t, a_t)$, where $s_1 = s$, and the average reward as $\rho(M, \pi, s) := \lim_{T \to \infty} \frac{1}{T} \mathbb{E}[R_T(M, \pi, s)]$, whenever the limit exists. In fact, the game value exists[2] [26]:

$$\rho^*(M, s) := \sup_{\pi^1} \inf_{\pi^2} \lim_{T \to \infty} \frac{1}{T} \mathbb{E}\left[R_T(M, \pi^1, \pi^2, s)\right].$$

If $\rho(M, \pi, s)$ or $\rho^*(M, s)$ does not depend on the initial state $s$, we simply write $\rho(M, \pi)$ or $\rho^*(M)$.

**The Bias Vector.** For a stationary policy $\pi$, the bias vector $h(M, \pi, \cdot)$ is defined, for each coordinate $s$, as

$$h(M, \pi, s) := \mathbb{E}\left[\sum_{t=1}^{\infty} r(s_t, a_t) - \rho(M, \pi, s) \Big| s_1 = s, a_t \sim \pi(\cdot|s_t)\right]. \tag{1}$$

The bias vector satisfies the Bellman equation: $\forall s \in \mathcal{S}$,

$$\rho(M, \pi, s) + h(M, \pi, s) = r(s, \pi) + \sum_{s'} p(s'|s, \pi) h(M, \pi, s'),$$

where $r(s, \pi) := \mathbb{E}_{a \sim \pi(\cdot|s)}[r(s, a)]$ and $p(s'|s, \pi) := \mathbb{E}_{a \sim \pi(\cdot|s)}[p(s'|s, a)]$.

The vector $h(M, \pi, \cdot)$ describes the relative *advantage* among states under model $M$ and (joint) policy $\pi$. The advantage (or disadvantage) of state $s$ compared to state $s'$ under policy $\pi$ is defined as the difference between the accumulated rewards with initial states $s$ and $s'$, which, from (1), converges to the difference $h(M, \pi, s) - h(M, \pi, s')$ asymptotically. For the ease of notation, the *span* of a vector $v$ is defined as $\mathrm{sp}(v) := \max_i v_i - \min_i v_i$. Therefore if a model, together with any policy, induces large $\mathrm{sp}(h)$, then this model will be difficult to learn because visiting a bad state costs a lot in the learning process. As shown in [3] for the MDP case, the regret has an inevitable dependency on $\mathrm{sp}(h(M, \pi^*, \cdot))$, where $\pi^*$ is the optimal policy.

On the other hand, $\mathrm{sp}(h(M, \pi, \cdot))$ is closely related to the mean first passage time under the Markov chain induced by $M$ and $\pi$. Actually we have $\mathrm{sp}(h(M, \pi, \cdot)) \leq \bar{T}^\pi(M) := \max_{s,s'} T_{s \to s'}^\pi(M)$, where $T_{s \to s'}^\pi(M)$ denotes the expected time to reach state $s'$ starting from $s$ when the model is $M$ and the player(s) follow the (joint) policy $\pi$. This fact is intuitive, and the proof can be seen at Remark M.1.

**Notations.** In order to save space, we often write equations in vector or matrix form. We use vectors inequalities: if $u, v \in \mathbb{R}^n$, then $u \leq v \Leftrightarrow u_i \leq v_i \ \forall i = 1, ..., n$. For a general matrix game with matrix $G$ of size $n \times m$, we denote the value of the game as $\mathrm{val}\, G := \max_{p \in \Delta_n} \min_{q \in \Delta_m} p^\top G q = \min_{q \in \Delta_m} \max_{p \in \Delta_n} p^\top G q$, where $\Delta_k$ is the probability simplex of dimension $k$. In SGs, given the estimated value function $u(s') \ \forall s'$, we often need to solve the following matrix game equation:

$$v(s) = \max_{a^1 \sim \pi^1(\cdot|s)} \min_{a^2 \sim \pi^2(\cdot|s)} \{r(s, a^1, a^2) + \sum_{s'} p(s'|s, a^1, a^2) u(s')\},$$

and this is abbreviated with the vector form $v = \mathrm{val}\{r + Pu\}$. We also use $\mathrm{solve}_1\, G$ and $\mathrm{solve}_2\, G$ to denote the optimal solutions of $p$ and $q$. In addition, the indicator function is denoted by $\mathbb{1}\{\cdot\}$ or $\mathbb{1}_{\{\cdot\}}$.

# 3 Problem Settings and Results Overview

We assume that the game proceeds for $T$ steps. In order to have meaningful regret bounds (i.e., sub-linear to $T$), we must make some assumptions to the SG model itself. Our two different assumptions are

**Assumption 1.** $\max\limits_{s,s'} \max\limits_{\pi^1 \in \Pi^{SR}} \max\limits_{\pi^2 \in \Pi^{SR}} T^{\pi^1,\pi^2}_{s \to s'}(M) \leq D.$

**Assumption 2.** $\max\limits_{s,s'} \max\limits_{\pi^2 \in \Pi^{SR}} \min\limits_{\pi^1 \in \Pi^{SR}} T^{\pi^1,\pi^2}_{s \to s'}(M) \leq D.$

Why we make these assumptions is as follows. Consider an SG model where the opponent (Player 2) has some way to lock the learner (Player 1) to some bad state. The best strategy for the learner might be to totally avoid, if possible, entering that state. However, in the early stage of the learning process, the learner won't know this, and he/she will have a certain probability to visit that state and get locked. This will cause linear regret to the learner. Therefore, we assume the following: whatever policy the opponent executes, the learner always has some way to reach any state within some bounded time. This is essentially our Assumption 2.

Assumption 1 is the stronger one that actually implies that under any policies executed by the players (not necessarily stationary, see Remark M.2), every state is visited within an average of $D$ steps. We find that under this assumption, the asymptotic regret can be improved. This assumption also has a sense similar to those required for Q-learning-type algorithms' convergence: they require that every state be visited infinitely often. See [18] for example.

These assumptions define some notion of *diameters* that are specific to the SG model. It is known that under Assumption 1 or Assumption 2, both players have optimal stationary policies, and the game value is independent of the initial state. Thus we can simply write $\rho^*(M,s)$ as $\rho^*(M)$. For a proof of these facts, please refer to Theorem E.1 in the appendix.

## 3.1 Two Settings and Results Overview

We focus on training Player 1 and discuss two settings. In the online setting, Player 1 competes with an arbitrary Player 2. The regret is defined as

$$\text{Reg}_T^{(on)} = \sum_{t=1}^{T} \rho^*(M) - r(s_t, a_t).$$

In the offline setting, we control both Player 1 and Player 2's actions, and find Player 1's maximin policy. The sample complexity is defined as

$$L_\varepsilon = \sum_{t=1}^{T} \mathbb{1}\{\rho^*(M) - \min_{\pi^2} \rho(M, \pi_t^1, \pi^2) > \varepsilon\},$$

where $\pi_t^1$ is a stationary policy being executed by Player 1 at time $t$. This definition is similar to those in [20, 19] for one-player MDPs. By the definition of $L_\varepsilon$, if we have an upper bound for $L_\varepsilon$ and run the algorithm for $T > L_\varepsilon$ steps, there is some $t$ such that $\pi_t^1$ is $\varepsilon$-optimal. We will explain how to pick this $t$ in Section 7 and Appendix L.

It turns out that we can use almost the same algorithm to handle these two settings. Since learning in the online setting is more challenging, from now on we will mainly focus on the online setting, and leave the discussion about the offline setting at the end of the paper. Our results can be summarized by the following two theorems.

**Theorem 3.1.** *Under Assumption 1, UCSG achieves* $Reg_T^{(on)} = \tilde{\mathcal{O}}(D^3 S^5 A + DS\sqrt{AT})$ *w.h.p.* [3]

**Theorem 3.2.** *Under Assumption 2, UCSG achieves* $Reg_T^{(on)} = \tilde{\mathcal{O}}(\sqrt[3]{DS^2AT^2})$ *w.h.p.*

# 4 Upper Confidence Stochastic Game Algorithm (UCSG)

**Algorithm 1** UCSG

---

**Input:** $\mathcal{S}, \mathcal{A} = \mathcal{A}^1 \times \mathcal{A}^2, T$.
**Initialization:** $t = 1$.
**for** phase $k = 1, 2, ...$ **do**
  $t_k = t$.
  **1. Initialize phase $k$:** $v_k(s, a) = 0, \quad n_k(s, a) = \max \left\{ 1, \sum_{\tau=1}^{t_k-1} \mathbb{1}_{(s_\tau, a_\tau)=(s,a)} \right\},$
  $n_k(s, a, s') = \sum_{\tau=1}^{t_k-1} \mathbb{1}_{(s_\tau, a_\tau, s_{\tau+1})=(s,a,s')}, \quad \hat{p}_k(s'|s, a) = \frac{n_k(s,a,s')}{n_k(s,a)}, \quad \forall s, a, s'.$
  **2. Update the confidence set:** $\mathcal{M}_k = \{ \tilde{M} : \forall s, a, \quad \tilde{p}(\cdot|s, a) \in \mathcal{P}_k(s, a) \}$, where
  $\mathcal{P}_k(s, a) := \text{CONF}_1(\hat{p}_k(\cdot|s, a), n_k(s, a)) \cap \text{CONF}_2(\hat{p}_k(\cdot|s, a), n_k(s, a)).$
  **3. Optimistic planning:** $\left( M_k^1, \pi_k^1 \right) = \text{MAXIMIN-EVI}\left( \mathcal{M}_k, \gamma_k \right),$ where $\gamma_k := 1/\sqrt{t_k}$.
  **4. Execute policies:**
    **repeat**
      Draw $a_t^1 \sim \pi_k^1(\cdot|s_t)$; observe the reward $r_t$ and the next state $s_{t+1}$.
      Set $v_k(s_t, a_t) = v_k(s_t, a_t) + 1$ and $t = t + 1$.
    **until** $\exists (s, a)$ such that $v_k(s, a) = n_k(s, a)$
**end for**
**Definitions of confidence regions:**
$\text{CONF}_1(\hat{p}, n) := \left\{ \tilde{p} \in [0, 1]^S : \|\tilde{p} - \hat{p}\|_1 \leq \sqrt{\frac{2S\ln(1/\delta_1)}{n}} \right\}, \quad \delta_1 = \frac{\delta}{2S^2 A \log_2 T}.$
$\text{CONF}_2(\hat{p}, n) := \left\{ \tilde{p} \in [0, 1]^S : \forall i, \left| \sqrt{\tilde{p}_i(1-\tilde{p}_i)} - \sqrt{\hat{p}_i(1-\hat{p}_i)} \right| \leq \sqrt{\frac{2\ln(6/\delta_1)}{n-1}}, \right.$
$\left. |\tilde{p}_i - \hat{p}_i| \leq \min\left( \sqrt{\frac{\ln(6/\delta_1)}{2n}}, \sqrt{\frac{2\hat{p}_i(1-\hat{p}_i)}{n} \ln\frac{6}{\delta_1}} + \frac{7}{3(n-1)}\ln\frac{6}{\delta_1} \right) \right\}.$

---

The Upper Confidence Stochastic Game algorithm (UCSG) (Algorithm 1) extends UCRL2 [19], using the optimism principle to balance exploitation and exploration. It proceeds in phases (indexed by $k$), and only changes the learner's policy $\pi_k^1$ at the beginning of each phase. The length of each phase is not fixed a priori, but depends on the statistics of past observations.

In the beginning of each phase $k$, the algorithm estimates the transition probabilities using empirical frequencies $\hat{p}_k(\cdot|s, a)$ observed in previous phases (Step 1). With these empirical frequencies, it can then create a confidence region $\mathcal{P}_k(s, a)$ for each transition probability. The transition probabilities lying in the confidence regions constitute a set of plausible stochastic game models $\mathcal{M}_k$, where the true model $M$ belongs to with high probability (Step 2). Then, Player 1 optimistically picks one model $M_k^1$ from $\mathcal{M}_k$, and finds the optimal (stationary) policy $\pi_k^1$ under this model (Step 3). Finally, Player 1 executes the policy $\pi_k^1$ for a while until some $(s, a)$-pair's number of occurrences is doubled during this phase (Step 4). The count $v_k(s, a)$ records the number of steps the $(s, a)$-pair is observed in phase $k$; it is reset to zero in the beginning of every phase.

In Step 3, to pick an optimistic model and a policy is to pick $M_k^1 \in \mathcal{M}_k$ and $\pi_k^1 \in \Pi^{\text{SR}}$ such that $\forall s$,

$$\min_{\pi^2} \rho(M_k^1, \pi_k^1, \pi^2, s) \geq \max_{\tilde{M} \in \mathcal{M}_k} \rho^*(\tilde{M}, s) - \gamma_k. \tag{2}$$

where $\gamma_k$ denotes the error parameter for MAXIMIN-EVI. The LHS of (2) is well-defined because Player 2 has stationary optimal policy under the MDP induced by $M_k^1$ and $\pi_k^1$. Roughly speaking, (2) says that $\min_{\pi^2} \rho(M_k^1, \pi_k^1, \pi^2, s)$ should approximate $\max_{\tilde{M} \in \mathcal{M}_k, \pi^1} \min_{\pi^2} \rho(\tilde{M}, \pi^1, \pi^2, s)$ by an error no more than $\gamma_k$. That is, $(M_k^1, \pi_k^1)$ are picked optimistically in $\mathcal{M}_k \times \Pi^{\text{SR}}$ considering the most adversarial opponent.

## 4.1 Extended SG and Maximin-EVI

The calculation of $M_k^1$ and $\pi_k^1$ involves the technique of Extended Value Iteration (EVI), which also appears in [19] as a one-player version.

Consider the following SG, named $M^+$. Let the state space $\mathcal{S}$ and Player 2's action space $\mathcal{A}^2$ remain the same as in $M$. Let $\mathcal{A}^{1+}, p^+(\cdot|\cdot, \cdot, \cdot), r^+(\cdot, \cdot, \cdot)$ be Player 1's action set, the transition kernel, and

the reward function of $M^+$, such that for any $a^1 \in \mathcal{A}^1$ and $a^2 \in \mathcal{A}^2$ and an admissible transition probability $\tilde{p}(\cdot|s, a^1, a^2) \in \mathcal{P}_k(s, a^1, a^2)$, there is an action $a^{1+} \in \mathcal{A}^{1+}$ such that $p^+(\cdot|s, a^{1+}, a^2) = \tilde{p}(\cdot|s, a^1, a^2)$ and $r^+(s, a^{1+}, a^2) = r(s, a^1, a^2)$. In other words, Player 1 selecting an action in $\mathcal{A}^{1+}$ is equivalent to selecting an action in $\mathcal{A}^1$ and simultaneously selecting an admissible transition probability in the confidence region $\mathcal{P}_k(\cdot, \cdot)$.

Suppose that $M \in \mathcal{M}_k$, then the extended SG $M^+$ satisfies Assumption 2 because the true model $M$ is embedded in $M^+$. By Theorem E.1 in Appendix E, it has a constant game value $\rho^*(M^+)$ independent of the initial state, and satisfies Bellman equation of the form $\text{val}\{r + Pf\} = \rho \cdot \mathbf{e} + f$, for some bounded function $f(\cdot)$, where $\mathbf{e}$ stands for the all-one constant vector. With the above conditions, we can use value iteration with Schweitzer transform (a.k.a. aperiodic transform)[34] to solve the optimal policy in the extended EG $M^+$. We call it MAXIMIN-EVI. For the details of MAXIMIN-EVI, please refer to Appendix F. We only summarize the result with the following Lemma.

**Lemma 4.1.** *Suppose the true model $M \in \mathcal{M}_k$, then the estimated model $M_k^1$ and stationary policy $\pi_k^1$ output by* MAXIMIN-EVI *in Step 3 satisfy*

$$\forall s, \quad \min_{\pi^2} \rho(M_k^1, \pi_k^1, \pi^2, s) \geq \max_{\pi^1} \min_{\pi^2} \rho(M, \pi^1, \pi^2, s) - \gamma_k.$$

Before diving into the analysis under the two assumptions, we first establish the following fact.

**Lemma 4.2.** *With high probability, the true model $M \in \mathcal{M}_k$ for all phases $k$.*

It is proved in Appendix D. With Lemma 4.2, we can fairly assume $M \in \mathcal{M}_k$ in most of our analysis.

## 5 Analysis under Assumption 1

In this section, we import analysis techniques from one-player MDPs [2, 19, 22, 9]. We also develop some techniques that deal with non-stationary opponents.

We model Player 2's behavior in the most general way, i.e., assuming it using a history-dependent randomized policy. Let $H_t = (s_1, a_1, r_1, ..., s_{t-1}, a_{t-1}, r_{t-1}, s_t) \in \mathcal{H}_t$ be the history up to $s_t$, then we assume $\pi_t^2$ to be a mapping from $\mathcal{H}_t$ to a distribution over $\mathcal{A}^2$. We will simply write $\pi_t^2(\cdot)$ and hide its dependency on $H_t$ inside the subscript $t$. A similar definition applies to $\pi_t^1(\cdot)$. With abuse of notations, we denote by $k(t)$ the phase where step $t$ lies in, and thus our algorithm uses policy $\pi_t^1(\cdot) = \pi_{k(t)}^1(\cdot|s_t)$. The notations $\pi_t^1$ and $\pi_k^1$ are used interchangeably. Let $T_k := t_{k+1} - t_k$ be the length of phase $k$. We decompose the regret in phase $k$ in the following way:

$$\Lambda_k := T_k \rho^*(M) - \sum_{t=t_k}^{t_{k+1}-1} r(s_t, a_t) = \sum_{n=1}^{4} \Lambda_k^{(n)}, \tag{3}$$

in which we define

$$\Lambda_k^{(1)} = T_k \left( \rho^*(M) - \min_{\pi^2} \rho(M_k^1, \pi_k^1, \pi^2, s_{t_k}) \right),$$

$$\Lambda_k^{(2)} = T_k \left( \min_{\pi^2} \rho(M_k^1, \pi_k^1, \pi^2, s_{t_k}) - \rho(M_k^1, \pi_k^1, \bar{\pi}_k^2, s_{t_k}) \right),$$

$$\Lambda_k^{(3)} = T_k \left( \rho(M_k^1, \pi_k^1, \bar{\pi}_k^2, s_{t_k}) - \rho(M, \pi_k^1, \bar{\pi}_k^2) \right),$$

$$\Lambda_k^{(4)} = T_k \rho(M, \pi_k^1, \bar{\pi}_k^2) - \sum_{t=t_k}^{t_{k+1}-1} r(s_t, a_t),$$

where $\bar{\pi}_k^2$ is some stationary policy of Player 2 which will be defined later. Since the actions of Player 2 are arbitrary, $\bar{\pi}_k^2$ is imaginary and only exists in analysis. Note that under Assumption 1, any stationary policy pair over $M$ induces an irreducible Markov chain, so we do not need to specify the initial states for $\rho(M, \pi_k^1, \bar{\pi}_k^2)$ in (3). Among the four terms, $\Lambda_k^{(2)}$ is clearly non-positive, and $\Lambda_k^{(1)}$, by optimism, can be bounded using Lemma 4.1. Now remains to bound $\Lambda_k^{(3)}$ and $\Lambda_k^{(4)}$.

## 5.1 Bounding $\sum_k \Lambda_k^{(3)}$ and $\sum_k \Lambda_k^{(4)}$

**The Introduction of $\bar{\pi}_k^2$.** $\Lambda_k^{(3)}$ and $\Lambda_k^{(4)}$ involve the artificial policy $\bar{\pi}_k^2$, which is a stationary policy that replaces Player 2's non-stationary policy in the analysis. This replacement costs some constant regret but facilitates the use of perturbation analysis in regret bounding. The selection of $\bar{\pi}_k^2$ is based on the principle that the behavior (e.g., total number of visits to some $(s, a)$) of the Markov chain induced by $M, \pi_k^1, \bar{\pi}_k^2$ should be close to the empirical statistics. Intuitively, $\bar{\pi}_k^2$ can be defined as

$$\bar{\pi}_k^2(a^2|s) := \frac{\sum_{t=t_k}^{t_{k+1}-1} \mathbb{1}_{s_t=s} \pi_t^2(a^2)}{\sum_{t=t_k}^{t_{k+1}-1} \mathbb{1}_{s_t=s}}. \tag{4}$$

Note two things, however. First, since we need the actual trajectory in defining this policy, it can only be defined *after* phase $k$ has ended. Second, $\bar{\pi}_k^2$ can be undefined because the denominator of (4) can be zero. However, this will not happen in too many steps. Actually, we have

**Lemma 5.1.** $\sum_k T_k \mathbb{1}\{\bar{\pi}_k^2 \text{ not well-defined}\} \le \tilde{\mathcal{O}}(DS^2A)$ *with high probability.*

Before describing how we bound the regret with the help of $\bar{\pi}_k^2$ and the perturbation analysis, we establish the following lemma:

**Lemma 5.2.** *We say the transition probability at time step $t$ is $\varepsilon$-accurate if $|p_k^1(s'|s_t, \pi_t) - p(s'|s_t, \pi_t)| \le \varepsilon \; \forall s'$ where $p_k^1$ denotes the transition kernel of $M_k^1$. We let $B_t(\varepsilon) = 1$ if the transition probability at time $t$ is $\varepsilon$-accurate; otherwise $B_t(\varepsilon) = 0$. Then for any state $s$, with high probability, $\sum_{t=1}^T \mathbb{1}_{s_t=s} \mathbb{1}_{B_t(\varepsilon)=0} \le \tilde{\mathcal{O}}\left(A/\varepsilon^2\right)$.*

Now we are able to sketch the logic behind our proofs. Let's assume that $\bar{\pi}_k^2$ models $\pi_k^2$ quite well, i.e., the expected frequency of every state-action pair induced by $M, \pi_k^1, \bar{\pi}_k^2$ is close to the empirical frequency induced by $M, \pi_k^1, \pi_k^2$. Then clearly, $\Lambda_k^{(4)}$ is close to zero in expectation. The term $\Lambda_k^{(3)}$ now becomes the difference of average reward between two Markov reward processes with slightly different transition probabilities. This term has a counterpart in [19] as a single-player version. Using similar analysis, we can prove that the dominant term of $\Lambda_k^{(3)}$ is proportional to $\mathrm{sp}(h(M_k^1, \pi_k^1, \bar{\pi}_k^2, \cdot))$. In the single-player case, [19] can directly claim that $\mathrm{sp}(h(M_k^1, \pi_k^1, \cdot)) \le D$ (see their Remark 8), but unfortunately, this is not the case in the two-player version. [4]

To continue, we resort to the perturbation analysis for the mean first passage times (developed in Appendix C). Lemma 5.2 shows that $M_k^1$ will not be far from $M$ for too many steps. Then Theorem C.9 in Appendix C tells that if $M_k^1$ are close enough to $M$, $T^{\pi_k^1, \bar{\pi}_k^2}(M_k^1)$ can be bounded by $2T^{\pi_k^1, \bar{\pi}_k^2}(M)$. As Remark M.1 implies that $\mathrm{sp}(h(M_k^1, \pi_k^1, \bar{\pi}_k^2, \cdot)) \le T^{\pi_k^1, \bar{\pi}_k^2}(M_k^1)$ and Assumption 1 guarantees that $T^{\pi_k^1, \bar{\pi}_k^2}(M) \le D$, we have $\mathrm{sp}(h(M_k^1, \pi_k^1, \bar{\pi}_k^2, \cdot)) \le T^{\pi_k^1, \bar{\pi}_k^2}(M_k^1) \le 2T^{\pi_k^1, \bar{\pi}_k^2}(M) \le 2D$.

The above approach leads to Lemma 5.3, which is a key in our analysis. We first define some notations. Under Assumption 1, any pair of stationary policies induces an irreducible Markov chain, which has a unique stationary distribution. If the policy pair $\pi = (\pi^1, \pi^2)$ is executed, we denote its stationary distribution by $\mu(M, \pi^1, \pi^2, \cdot) = \mu(M, \pi, \cdot)$. Besides, denote $v_k(s) := \sum_{t=t_k}^{t_{k+1}-1} \mathbb{1}_{s_t=s}$.

We say a phase $k$ is benign if the following hold true: the true model $M$ lies in $\mathcal{M}_k$, $\bar{\pi}_k^2$ is well-defined, $\mathrm{sp}(h(M_k^1, \pi_k^1, \bar{\pi}_k^2, \cdot)) \le 2D$, and $\mu(M, \pi_k^1, \bar{\pi}_k^2, s) \le \frac{2v_k(s)}{T_k} \; \forall s$. We can show the following:

**Lemma 5.3.** $\sum_k T_k \mathbb{1}\{\text{phase } k \text{ is not benign}\} \le \tilde{\mathcal{O}}(D^3 S^5 A)$ *with high probability.*

Finally, for benign phases, we can have the following two lemmas.

**Lemma 5.4.** $\sum_k \Lambda_k^{(4)} \mathbb{1}\{\bar{\pi}_k^2 \text{ is well-defined }\} \le \tilde{\mathcal{O}}(D\sqrt{ST} + DSA)$ *with high probability.*

**Lemma 5.5.** $\sum_k \Lambda_k^{(3)} \mathbb{1}\{$*phase $k$ is benign*$\} \leq \tilde{\mathcal{O}}(DS\sqrt{AT} + DS^2A)$ *with high probability,*

*Proof of Theorem 3.1.* The regret proof starts from the decomposition of (3). $\Lambda_k^{(1)}$ is bounded with the help of Lemma 4.1: $\sum_k \Lambda_k^{(1)} \leq \sum_k T_k/\sqrt{t_k} = \mathcal{O}(\sqrt{T})$. $\sum_k \Lambda_k^{(2)} \leq 0$ by definition. Then with Lemma 5.1, 5.3, 5.4, and 5.5, we can bound $\Lambda_k^{(3)}$ and $\Lambda_k^{(4)}$ by $\tilde{\mathcal{O}}(D^3S^5A + DS\sqrt{AT})$. $\qquad\square$

# 6 Analysis under Assumption 2

In Section 5, the main ingredient of regret analysis lies in bounding the span of the bias vector, $\mathrm{sp}(h(M_k^1, \pi_k^1, \bar{\pi}_k^2, \cdot))$. However, the same approach does not work because under the weaker Assumption 2, we do not have a bound on the mean first passage time under arbitrary policy pairs. Hence we adopt the approach of approximating the average reward SG problem by a sequence of finite-horizon SGs: on a high level, first, with the help of Assumption 2, we approximate the $T$ multiple of the original average-reward SG game value (i.e. the total reward in hindsight) with the sum of those of $H$-step episodic SGs; second, we resort to [9]'s results to bound the $H$-step SGs' sample complexity and translates it to regret.

**Approximation by repeated episodic SGs.** For the approximation, the quantity $H$ does not appear in UCSG but only in the analysis. The horizon $T$ is divided into episodes each with length $H$. Index episodes with $i = 1, ..., T/H$, and denote episode $i$'s first time step by $\tau_i$. We say $i \in \mathrm{ph}(k)$ if all $H$ steps of episode $i$ lie in phase $k$. Define the $H$-step expected reward under joint policy $\pi$ with initial state $s$ as $V_H(M, \pi, s) := \mathbb{E}\left[\sum_{t=1}^H r_t | a_t \sim \pi, s_1 = s\right]$. Now we decompose the regret in phase $k$ as

$$\Delta_k := T_k \rho^* - \sum_{t=t_k}^{t_{k+1}-1} r(s_t, a_t) \leq \sum_{n=1}^6 \Delta_k^{(n)}, \tag{5}$$

where

$$\Delta_k^{(1)} = \sum_{i \in \mathrm{ph}(k)} H\left(\rho^* - \min_{\pi^2} \rho(M_k^1, \pi_k^1, \pi^2, s_{\tau_i})\right),$$

$$\Delta_k^{(2)} = \sum_{i \in \mathrm{ph}(k)} \left(H \min_{\pi^2} \rho(M_k^1, \pi_k^1, \pi^2, s_{\tau_i}) - \min_{\pi^2} V_H(M_k^1, \pi_k^1, \pi^2, s_{\tau_i})\right),$$

$$\Delta_k^{(3)} = \sum_{i \in \mathrm{ph}(k)} \left(\min_{\pi^2} V_H(M_k^1, \pi_k^1, \pi^2, s_{\tau_i}) - V_H(M_k^1, \pi_k^1, \pi_i^2, s_{\tau_i})\right),$$

$$\Delta_k^{(4)} = \sum_{i \in \mathrm{ph}(k)} \left(V_H(M_k^1, \pi_k^1, \pi_i^2, s_{\tau_i}) - V_H(M, \pi_k^1, \pi_i^2, s_{\tau_i})\right),$$

$$\Delta_k^{(5)} = \sum_{i \in \mathrm{ph}(k)} \left(V_H(M, \pi_k^1, \pi_i^2, s_{\tau_i}) - \sum_{t=\tau_i}^{\tau_{i+1}-1} r(s_t, a_t)\right), \quad \Delta_k^{(6)} = 2H.$$

Here, $\pi_i^2$ denotes Player 2's policy in episode $i$, which may be non-stationary. $\Delta_k^{(6)}$ comes from the possible two incomplete episodes in phase $k$. $\Delta_k^{(1)}$ is related to the tolerance level we set for the MAXIMIN-EVI algorithm: $\Delta_k^{(1)} \leq T_k\gamma_k = T_k/\sqrt{t_k}$. $\Delta_k^{(2)}$ is an error caused by approximating an infinite-horizon SG by a repeated episodic $H$-step SG (with possibly different initial states). $\Delta_k^{(3)}$ is clearly non-positive. It remains to bound $\Delta_k^{(2)}, \Delta_k^{(4)}$ and $\Delta_k^{(5)}$.

**Lemma 6.1.** *By Azuma-Hoeffding's inequality, $\sum_k \Delta_k^{(5)} \leq \tilde{\mathcal{O}}(\sqrt{HT})$ with high probability.*

**Lemma 6.2.** *Under Assumption 2, $\sum_k \Delta_k^{(2)} \leq TD/H + \sum_k T_k\gamma_k$.*

**From sample complexity to regret bound.** As the main contributor of regret, $\Delta_k^{(4)}$ corresponds to the inaccuracy in the transition probability estimation. Here we largely reuse [9]'s results where they consider one-player episodic MDP with a fixed initial state distribution. Their main lemma states that the number of episodes in phases such that $|V_H(M_k^1, \pi_k, s_0) - V_H(M, \pi_k, s_0)| > \varepsilon$ will not exceed $\tilde{\mathcal{O}}\left(H^2S^2A/\varepsilon^2\right)$, where $s_0$ is their initial state in each episode. In other words, $\sum_k \frac{T_k}{H} \mathbb{1}\{|V_H(M_k^1, \pi_k, s_0) - V_H(M, \pi_k, s_0)| > \varepsilon\} = \tilde{\mathcal{O}}(H^2S^2A/\varepsilon^2)$. Note that their proof allows $\pi_k$ to be an arbitrarily selected non-stationary policy for phase $k$.

We can directly utilize their analysis and we summarize it as Theorem K.1 in the appendix. While their algorithm has an input $\varepsilon$, this input can be removed without affecting bounds. This means that the PAC bounds holds for arbitrarily selected $\varepsilon$. With the help of Theorem K.1, we have

**Lemma 6.3.** $\sum_k \Delta_k^{(4)} \leq \tilde{\mathcal{O}}(S\sqrt{HAT} + HS^2A)$ *with high probability.*

*Proof of Theorem 3.2.* With the decomposition (5) and the help of Lemma 6.1, 6.2, and 6.3, the regret is bounded by $\tilde{O}(\frac{TD}{H} + S\sqrt{HAT} + S^2AH) = \tilde{\mathcal{O}}(\sqrt[3]{DS^2AT^2})$ by selecting $H = \max\{D, \sqrt[3]{D^2T/(S^2A)}\}$. □

# 7  Sample Complexity of Offline Training

In Section 3.1, we defined $L_\varepsilon$ to be the sample complexity of Player 1's maximin policy. In our offline version of UCSG, in each phase $k$ we let both players each select their own optimistic policy. After Player 1 has optimistically selected $\pi_k^1$, Player 2 then optimistically selects his policy $\pi_k^2$ based on the known $\pi_k^1$. Specifically, the model-policy pair $\left(M_k^2, \pi_k^2\right)$ is obtained by another extended value iteration on the extended MDP under fixed $\pi_k^1$, where Player 2's action set is extended. By setting the stopping threshold also as $\gamma_k$, we have

$$\rho(M_k^2, \pi_k^1, \pi_k^2, s) \leq \min_{\tilde{M} \in \mathcal{M}_k} \min_{\pi^2} \rho(\tilde{M}, \pi_k^1, \pi^2, s) + \gamma_k \tag{6}$$

when value iteration halts. With this selection rule, we are able to obtain the following theorems.

**Theorem 7.1.** *Under Assumption 1,* UCSG *achieves* $L_\varepsilon = \tilde{\mathcal{O}}(D^3S^5A + D^2S^2A/\varepsilon^2)$ *w.h.p.*

**Theorem 7.2.** *Let Assumption 2 hold, and further assume that* $\max\limits_{s,s'} \max\limits_{\pi^1 \in \Pi^{SR}} \min\limits_{\pi^2 \in \Pi^{SR}} T_{s \to s'}^{\pi^1, \pi^2}(M) \leq D$. *Then* UCSG *achieves* $L_\varepsilon = \tilde{\mathcal{O}}(DS^2A/\varepsilon^3)$ *w.h.p.*

The algorithm can output a single stationary policy for Player 1 with the following guarantee: if we run the offline version of UCSG for $T > L_\varepsilon$ steps, the algorithm can output a single stationary policy that is $\varepsilon$-optimal. We show how to output this policy in the proofs of Theorem 7.1 and 7.2.

# 8  Open Problems

In this work, we obtain the regret of $\tilde{\mathcal{O}}(D^3S^5A + DS\sqrt{AT})$ and $\tilde{\mathcal{O}}(\sqrt[3]{DS^2AT})$ under different mixing assumptions. A natural open problem is how to improve these bounds on both asymptotic and constant terms. A lower bound of them can be inherited from the one-player MDP setting, which is $\Omega(\sqrt{DSAT})$ [19].

Another open problem is that if we further weaken the assumptions to $\max_{s,s'} \min_{\pi_1} \min_{\pi_2} T_{s \to s'}^{\pi^1, \pi^2} \leq D$, can we still learn the SG? We have argued that if we only have this assumption, in general we cannot get sublinear regret in the online setting. However, it is still possible to obtain polynomial-time offline sample complexity if the two players cooperate to explore the state-action space.

## Acknowledgments

We would like to thank all anonymous reviewers who have devoted their time for reviewing this work and giving us valuable feedbacks. We would like to give special thanks to the reviewer who reviewed this work's previous version in ICML; your detailed check of our proofs greatly improved the quality of this paper.

## Footnotes

[1]Turn-based SGs, like Go, are special cases: in each state, one player's action set contains only a null action.

[2]Unlike in one-player MDPs, the $\sup$ and $\inf$ in the definition of $\rho^*(M, s)$ are not necessarily attainable. Moreover, players may not have stationary optimal policies.

[3]We write, "with high probability, $g = \tilde{\mathcal{O}}(f)$" or "*w.h.p.*, $g = \tilde{\mathcal{O}}(f)$" to indicate "with probability $\geq 1 - \delta$, $g = f_1 \mathcal{O}(f) + f_2$", where $f_1, f_2$ are some polynomials of $\log D, \log S, \log A, \log T, \log(1/\delta)$.

[4]The argument in [19] is simple: suppose that $h(M_k^1, \pi_k^1, s) - h(M_k^1, \pi_k^1, s') > D$, by the communicating assumption, there is a path from $s'$ to $s$ with expected time no more than $D$. Thus a policy that first goes from $s'$ to $s$ within $D$ steps and then executes $\pi_k^1$ will outperform $\pi_k^1$ at $s'$. This leads to a contradiction. In two-player SGs, with a similar argument, we can also show that $\mathrm{sp}(h(M_k^1, \pi_k^1, \pi_k^{2*}, \cdot)) \le D$, where $\pi_k^{2*}$ is the best response to $\pi_k^1$ under $M_k^1$. However, since Player 2 is uncontrollable, his/her policy $\pi_k^2$ (or $\bar{\pi}_k^2$) can be quite different from $\pi_k^{2*}$, and thus $\mathrm{sp}(h(M_k^1, \pi_k^1, \bar{\pi}_k^2, \cdot)) \le D$ does not necessarily hold true.

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
