[Supplementary Material]

# Contents (Appendix)

## A Previous Bounds for MDPs and SGs

The techniques we use in this paper are most related to the probably approximately correct (PAC) analysis for RL algorithms. Some rather complete reviews of the related works are provided in [19, 9]. [19] considers the average-reward MDP that is *communicating* with bounded diameter $D$ (i.e., $\max_{s,s'} \min_\pi T^\pi_{s \to s'}(M) \leq D$, where $T^\pi_{s \to s'}(M)$ is defined as the expected time to reach from state $s$ to state $s'$ under model $M$ and policy $\pi$). Their UCRL2 algorithm achieves $\tilde{\mathcal{O}}(DS\sqrt{AT})$ regret upper bound, while still having a gap with the $\Omega(\sqrt{DSAT})$ lower bound. These bounds translate to $\tilde{\mathcal{O}}\left(\frac{D^2 S^2 A}{\varepsilon^2}\right)$ and $\Omega\left(\frac{DSA}{\varepsilon^2}\right)$ sample complexity. The additional $D$ dependency is resolved by [22, 9], though in discounted and episodic settings respectively. These two works leverage the Bellman equation for local variance and obtained sample complexity bounds of order $\tilde{\mathcal{O}}\left(\frac{S^2 A}{\varepsilon^2 (1-\gamma)^3}\right)$ and $\tilde{\mathcal{O}}\left(\frac{H^2 S^2 A}{\varepsilon^2}\right)$ ($\gamma$: discount factor, $H$: fixed horizon length), making their gaps with the lower bounds $\Omega\left(\frac{SA}{\varepsilon^2 (1-\gamma)^3}\right)$ and $\Omega\left(\frac{H^2 SA}{\varepsilon^2}\right)$ remain only an order of $S$.

The scenario that most resembles ours in the literature is that considered in [5], who proposed the algorithm R-MAX. R-MAX is an optimism-based algorithm that can be used to learn stochastic games with arbitrary opponents. However, the algorithm depends on a parameter $\varepsilon$ and the $\varepsilon$-return mixing time $T_\varepsilon$ that need to be known in advance. This $\varepsilon$-return mixing time resembles our $\frac{D}{\varepsilon}$ in Assumption 2. As a result, their $\tilde{\mathcal{O}}\left(\frac{T_\varepsilon^3 S^2 A}{\varepsilon^3}\right)$ translates to $\tilde{\mathcal{O}}\left(\frac{D^3 S^2 A}{\varepsilon^6}\right)$, while our bound is $\tilde{\mathcal{O}}\left(\frac{DS^2 A}{\varepsilon^3}\right)$. Another

difference lies in that the output policy of our algorithm is a stationary one, rather than a $T_\varepsilon$-step non-stationary policy as in R-MAX.

## B   Inequalities

**Lemma B.1.** *(Azuma-Hoeffding's inequality. Theorem 4.2 of [6]) Let $\mathcal{F}_1 \subseteq \cdots \subseteq \mathcal{F}_T$ be a filtration, and $Y_1, \cdots, Y_T$ real random variables such that $Y_t$ is $\mathcal{F}_t$-measurable, $\mathbb{E}(Y_t|\mathcal{F}_{t-1}) = 0$ and $Y_t \in [A_t, A_t + c_t]$ where $A_t$ is a random variable $\mathcal{F}_{t-1}$-measurable and $c_t$ is a positive constant. Then with probability at least $1 - \delta$,*

$$\sum_{t=1}^{T} Y_t < \sqrt{\frac{\log(\delta^{-1})}{2} \sum_{t=1}^{T} c_t^2}.$$

**Lemma B.2.** *(Bernstein inequality. Lemma 4.4 of [6]) Let $\mathcal{F}_1 \subseteq \cdots \subseteq \mathcal{F}_T$ be a filtration, and $Y_1, \cdots, Y_T$ real random variables such that $Y_t$ is $\mathcal{F}_t$-measurable, $\mathbb{E}(Y_t|\mathcal{F}_{t-1}) = 0$ and $|Y_t| \leq b$ for some $b > 0$. Let $V_T = \sum_{t=1}^{T} \mathbb{E}(Y_t^2|\mathcal{F}_{t-1})$ and $\delta > 0$. Then with probability at least $1 - \delta$,*

$$\sum_{t=1}^{T} Y_t \leq 2\sqrt{V_T \log(T\delta^{-1})} + \sqrt{5}b \log(T\delta^{-1}).$$

## C   Perturbation Bounds for Markov Chains

Perturbation analysis for Markov chains plays an important role in analyzing reinforcement learning algorithms (e.g., [2]). Those analyses mainly center around the question that when the transition probabilities of a Markov chain are perturbed by a little, how much stationary distributions or mean first passage times (as defined in Definition C.1) will change. While in [2], the perturbation bound for stationary distributions is used, we further use that of the mean first passage time to get a tighter regret bound.

In this section, we use $i, j$ to index states, and use $\mu_i$ to denote the stationary distribution of state $i$ in an irreducible Markov chain.

**Definition C.1** (Mean first passage time). *In a Markov chain, we define $T_{ij}$ to be the expected time to reach state $j$ starting from state $i$. In the case $i = j$, $T_{ii}$ is the expected time to return to state $i$ when starting from $i$. Thus $T_{ij} \geq 1$ always holds whether $i = j$ or not.*

### C.1   Perturbation Bounds for Stationary Distribution

**Theorem C.2** (Proposition 2.2 of [7]). *Let $C$ and $\tilde{C}$ be two irreducible Markov chains with the same state space $\mathcal{S}$. Let their transition matrices be $P$, $\tilde{P}$, and stationary distributions be $\mu$, $\tilde{\mu}$. Let $E = \tilde{P} - P$ and use $\|\cdot\|_\infty$ to represent the largest absolute value in a matrix, then $\forall j$,*

$$|\tilde{\mu}_j - \mu_j| \leq \mu_j \frac{S\|E\|_\infty}{2} \max_{i \neq j} T_{ij}. \tag{7}$$

With a little modification on the proof of Theorem C.2, we can actually have the following lemma, which only requires that $C$ be an irreducible Markov chain.

**Theorem C.3.** *Let $C$ be an irreducible Markov chain, and $\tilde{C}$ be some Markov chain with the same state space $\mathcal{S}$ as $C$. Let their transition matrices be $P$, $\tilde{P}$, and let $C$'s stationary distributions be $\mu$. Let $E = \tilde{P} - P$. If $\|E\|_\infty < 2/(S \max_{i \neq j} T_{ij})$, then $\tilde{C}$ is also an irreducible Markov chain; furthermore, the stationary distribution of $\tilde{C}$, $\tilde{\mu}$, satisfies $\forall j$,*

$$|\tilde{\mu}_j - \mu_j| \leq \mu_j \frac{S\|E\|_\infty}{2} \max_{i \neq j} T_{ij}. \tag{8}$$

*Proof.* Let $P^*$ and $\tilde{P}^*$ be the Cesaro limits of $P$ and $\tilde{P}$, which is defined by $P^* = \lim_{T \to \infty} \frac{1}{T} \sum_{t=1}^{T} P^{t-1}$. Then we have

$$P^*(I - P) = 0, \quad \tilde{P}^*(I - \tilde{P}) = \tilde{P}^*(I - P - E) = 0,$$

and thus $(\tilde{P}^* - P^*)(I - P) = \tilde{P}^* E$. If $\tilde{P}$ induces an irreducible Markov chain, $\tilde{P}^*$ will have all identical rows and all positive elements. Suppose not, we can still extract its $k$-th row, which corresponds to the stationary distribution when starting from state $k$. Let this $k$-th row's $j$-th element be $\tilde{\mu}_j^k$. We can write $(\tilde{\mu}_j^k - \mu_j)(I - P) = \tilde{\mu}_j^k E$. Then following the same proof as in [7] or by [16]'s Theorem 2.1, we still have

$$|\tilde{\mu}_j^k - \mu_j| \le \mu_j \frac{S\,\|E\|_\infty}{2} \max_{i \ne j} T_{ij}$$

Now since $\|E\|_\infty \le 2/(S \max_{i \ne j} T_{ij})$ and $\mu_j > 0 \,\forall j$, we have $\tilde{\mu}_j^k > 0 \,\forall j, k$. This means that every state is recurrent and reachable from each other, implying that $\tilde{P}$ induces an irreducible Markov chain. $\qquad\square$

## C.2 Perturbation Bounds for Mean First Passage Time

The main result of this subsection is stated in Theorem C.9. It is developed with the help of Theorem C.5 to Theorem C.8.

**Definition C.4** (g-inverse, Definition 3.1 of [15]). *A g-inverse of a matrix $A$ is any matrix $G$ such that $AGA = A$.*

**Theorem C.5** (Theorem 5.3 of [16]). *Let $C$ be an irreducible Markov chain with stochastic matrix $P$. Let $T_{ij}$ be the first passage time from state $i$ to state $j$, and let $G$ be any g-inverse of $I - P$. We have*

$$\mu_j T_{ij} = G_{jj} - G_{ij} + \delta_{ij} + \mu_j \sum_{k=1}^n (G_{ik} - G_{jk}).$$

The below theorem introduces a special g-inverse that is convenient for our use.

**Theorem C.6** (Theorem 3.3 of [15]). *Let $P$ be a stochastic matrix of an irreducible Markov chain. Let $p_n^\top$ denote the $n$-th row of $P$, and $e_n$ denote the unit column vector with $n$-th component being 1. Then $I - P + e_n p_n^\top$ is non-singular, and $G = (I - P + e_n p_n^\top)^{-1}$ is a g-inverse of $I - P$.*

**Theorem C.7** (Section 5 of [16]). *Let $\tilde{P}$ be a stochastic matrix of an irreducible Markov chain perturbed from another stochastic matrix $P$ of an irreducible Markov chain. Suppose that the perturbation only occurs at the $n$-th row of $P$ (i.e. $p_i^\top = \tilde{p}_i^\top \,\forall i \ne n$). Define $G$ as that in Theorem C.6. Then $G = \tilde{G}$.*

Suppose that the perturbation only occurs at the $n$-th row, and let $G = (I - P + e_n p_n^\top)^{-1}$. Then Theorem C.5 and C.7 together imply that for $i \ne j$,

$$\tilde{T}_{ij} = T_{ij} + (G_{ij} - G_{jj})\left(\frac{1}{\mu_j} - \frac{1}{\tilde{\mu}_j}\right), \tag{9}$$

with $T_{jj} = 1/\mu_j$ and $\tilde{T}_{jj} = 1/\tilde{\mu}_j$ (Corollary 5.3.1 of [16]). Here we see that $\tilde{T}_{in} = T_{in}, \forall i$.

**Lemma C.8.** *Let $P$ be the stochastic matrix of an irreducible Markov chain, and let $G = (I - P + e_n p_n^\top)^{-1}$. If all mean first passage times are bounded by $D'$ (i.e., $T_{ij} \le D' \,\,\forall i, j$), then $|G_{ij} - G_{jj}| \le 2\mu_j D' \,\,\forall i, j$.*

*Proof.* We first verify that

$$G = \begin{bmatrix} (I - P_n)^{-1} & \mathbf{e} \\ 0 & 1 \end{bmatrix}, \tag{10}$$

where $P_n$ is obtained by deleting the $n$-th row and $n$-th column of $P$ (without loss of generality, assume that the $n$-th row is last row of $G$).

Directly expanding $I - P + e_n p_n^\top$, we get

$$I - P + e_n p_n^\top = \begin{bmatrix} (I - P_n) & d \\ 0 & 1 \end{bmatrix},$$

where $d = (-p_{1,n}, -p_{2,n}, ..., -p_{n-1,n})^\top$. To verify that $(I - P + e_n p_n^\top)^{-1}$ takes the form of (10), one only needs to verify that $(I - P_n)\mathbf{e} + d = 0$. This can be seen by $(I - P_n)\mathbf{e} = \mathbf{e} - P_n\mathbf{e} = (1, ..., 1) - (\sum_{i=1}^{n-1} p_{1,i}, ..., \sum_{i=1}^{n-1} p_{n-1,i}) = -d$.

For $i \neq n$, from $G$'s expression in (10), we have

$$\sum_{k=1}^{n} G_{ik} = e_i^\top G\mathbf{e} = e_i^\top (I - P_n)^{-1}\mathbf{e} + 1. \tag{11}$$

Note that the dimension of $e_i$ and $\mathbf{e}$ are $n$ in the second expression of (11), while are $n - 1$ in the third expression. By [7]'s Equation (2.3), $e_i^\top (I - P_n)^{-1}\mathbf{e} = T_{in}$. One can also see this by observing that $(I - P_n)^{-1} = I + P_n + P_n^2 + \cdots$, and $e_i^\top P_n^m e_j$ is "the probability of staying at $j$ after $m$ steps from $i$, while not visiting $n$ in any of the $m$ steps". Summing $e_i^\top P_n^m e_j$ over $j$ and $m$, the physical meaning becomes the mean first passage time from $i$ to $n$, and the mathematical expression becomes $e_i^\top (I - P_n)^{-1}\mathbf{e}$. Thus, $|\sum_{k=1}^{n}(G_{ik} - G_{jk})| = |T_{in} - T_{jn}| \leq \max_{ij} T_{ij} \leq D'$. By Theorem C.5, whenever $i \neq j$,

$$|G_{ij} - G_{jj}| = \left|\mu_j T_{ij} - \mu_j \sum_{k=1}^{n}(G_{ik} - G_{jk})\right| \leq \mu_j T_{ij} + \mu_j D' \leq 2\mu_j D'.$$

$\square$

We now combine (9) with (8) and Lemma C.8. Assuming that $T_{ij} \leq D'$, we have for $i \neq j$,

$$|\tilde{T}_{ij} - T_{ij}| = |G_{ij} - G_{jj}| \frac{|\tilde{\mu}_j - \mu_j|}{\mu_j \tilde{\mu}_j} \leq \frac{\mu_j}{\tilde{\mu}_j} \|E\|_\infty SD'^2. \tag{12}$$

With (8) and (12) available, we now consider a general perturbation, which can actually be decomposed as $S$ single-row perturbations.

**Theorem C.9.** *Let $P$, $\tilde{P}$ be the original and the perturbed stochastic matrices, and let $\{T_{ij}\}$, $\{\tilde{T}_{ij}\}$ be their corresponding mean first passage times. If $\max_{ij} T_{ij} \leq D$ and $\|E\|_\infty = \left\|\tilde{P} - P\right\|_\infty \leq \frac{1}{8DS^2}$, then $\max_{ij} \tilde{T}_{ij} \leq 2D$.*

*Proof.* We do this general perturbation of $P$ by perturbing one row at a time. This procedure will repeat for $S$ times.

Suppose that the original stationary distribution and first passage times are denoted by $\mu_i^{(0)}$ and $T_{ij}^{(0)}$, and that those after $n$-th perturbation are denoted by $\tilde{\mu}_i^{(n)}$ and $\tilde{T}_{ij}^{(n)}$.

Suppose that $T_{ij}^{(0)} \leq D$ $\forall i, j$ and $\mu_j^{(0)} \geq \frac{1}{D}$ $\forall j$. Set $\|E\|_\infty \leq \frac{1}{8S^2 D}$. We prove the following facts by induction:

$$\tilde{T}_{ij}^{(n)} \leq D\left(1 + \frac{n}{S}\right), \tag{13}$$

$$\tilde{\mu}_j^{(n)} \in \left[\tilde{\mu}_j^{(0)}\left(1 - \frac{1}{8S}\right)^n, \tilde{\mu}_j^{(0)}\left(1 + \frac{1}{8S}\right)^n\right], \tag{14}$$

for $n = 1, ..., S$. Since $n \leq S$, these induction hypotheses implicitly imply that

$$\tilde{T}_{ij}^{(n)} \leq 2D, \tag{15}$$

$$\tilde{\mu}_j^{(n)} \geq \tilde{\mu}_j^{(0)}\left(1 - \frac{1}{8S}\right)^S \geq \frac{1}{2D}, \tag{16}$$

because $(1 - 1/(8S))^S \geq 1/2$ for all $S \geq 1$. Now we start the induction. The base case for $n = 0$ clearly holds. Suppose that (13)-(14) hold for all $n \leq k$. Then by (8) we have $\left|\tilde{\mu}_j^{(k)} - \tilde{\mu}_j^{(k+1)}\right| \leq \tilde{\mu}_j^{(k)} \frac{1}{8S}$, so $\tilde{\mu}_j^{(k+1)} \geq \tilde{\mu}_j^{(k)}\left(1 - \frac{1}{8S}\right) \geq \tilde{\mu}_j^{(0)}\left(1 - \frac{1}{8S}\right)^{(k+1)}$, and $\tilde{\mu}_j^{(k+1)} \leq$

$\tilde{\mu}_j^{(k)}\left(1+\frac{1}{8S}\right) \le \tilde{\mu}_j^{(0)}\left(1+\frac{1}{8S}\right)^{(k+1)}$. On the other hand, by (12), for $i \ne j$, we have $\tilde{T}_{ij}^{(k+1)} - \tilde{T}_{ij}^{(k)} \le \frac{\tilde{\mu}_j^{(k)}}{\tilde{\mu}_j^{(k+1)}}\|E\|_\infty (2D)^2 \le \frac{1}{1-\frac{1}{8S}}\frac{1}{8S^2 D}S(2D)^2 \le \frac{D}{S}$, so $\tilde{T}_{ij}^{(k+1)} \le \tilde{T}_{ij}^{(k)} + \frac{D}{S} \le D\left(1+\frac{k+1}{S}\right)$. In the case $i=j$, we have $\tilde{T}_{jj}^{(k+1)} - \tilde{T}_{jj}^{(k)} = \frac{1}{\tilde{\mu}_j^{(k+1)}} - \frac{1}{\tilde{\mu}_j^{(k)}} \le \frac{\left(1+\frac{1}{8S}\right)-1}{\tilde{\mu}_j^{(k)}} \le \frac{2D}{8S} \le \frac{D}{S}$. $\square$

## D  Lemmas for Failing Events

**Lemma D.1** (Proposition 18 of [19]). *The number of phases is upper bounded by $U_{\max} = SA \log_2 T$.*

*Proof.* Since phase changes only occur when the sample count of some $(s, a^1, a^2)$ is doubled, those changes corresponding to a specific $(s, a^1, a^2)$ is upper bounded by $\log_2 T$. Considering all states and actions, the total number of phase changes is upper bounded by $SA \log_2 T$. $\square$

**Lemma D.2** (Lemma 17 of [19]). *For some specific $k, s$ and $a$, the event $p(\cdot|s,a) \in \text{CONF}_1(\hat{p}_k(\cdot|s,a), n_k(s,a))$ holds with probability at least $1 - \delta_1$.*

*Proof.* Please refer to [19]. $\square$

**Lemma D.3** (Lemma 1 of [9], Theorem 10 and 11 of [25]). *For some specific $k, s$ and $a$, the event $p(\cdot|s,a) \in \text{CONF}_2(\hat{p}_k(\cdot|s,a), n_k(s,a))$ holds with probability at least $1 - S\delta_1$.*

*Proof.* Please refer to [9] or [25]. $\square$

*Proof of Lemma 4.2.* By Lemma D.1, there are at most $SA \log_2 T$ confidence set updates to consider. Each update involves only a specific $\hat{p}(\cdot|s,a)$ (totally $S$ entries). By Lemma D.2, D.3 and using the union bound, the event $M \in \mathcal{M}_k \forall k$ holds with probability at least $1 - SA \log_2 T \times (\delta_1 + S\delta_1) \ge 1 - \delta$. $\square$

## E  Lemmas for Stationary Optimal Policies

**Theorem E.1.** *Given a stochastic game $M = (\mathcal{S}, \mathcal{A}, r, p)$, where $\mathcal{S}$ is countable, $\mathcal{A} = \mathcal{A}^1 \times \mathcal{A}^2$ a compact metric space and both $r(s, \cdot) \in [0, 1]$ and $p(s'|s, \cdot)$ are continuous in $a = (a^1, a^2)$. Suppose Assumption 2 holds for $M$. Then there exist maximin stationary policies $\pi^* = (\pi^{1*}, \pi^{2*})$ for the two-player zero-sum stochastic game, the maximin stationary policies attain the game value $\rho^*$, which is independent of the initial state, and there is a bounded function $h(\cdot)$ which together with $\rho^*$ satisfies the following Bellman equation. That is, for all state $s$,*

$$\rho^* + h(s) = \max_{\pi^1 \in \Pi^{SR}}\left\{r(s, \pi^1, \pi^{2*}) + \sum_{s'} p(s'|s, \pi^1, \pi^{2*})h(s')\right\}$$

$$\rho^* + h(s) = \min_{\pi^2 \in \Pi^{SR}}\left\{r(s, \pi^{1*}, \pi^2) + \sum_{s'} p(s'|s, \pi^{1*}, \pi^2)h(s')\right\}.$$

To prove this, we use the following lemma which connects the boundedness of mean first passage times with the uniform boundedness of $\text{sp}(V_\alpha^*(\cdot))$ for all discount factor $0 < \alpha < 1$, where $V_\alpha^*(\cdot)$ is the discounted game value defined as $V_\alpha^*(s) = \max_{\pi^1}\min_{\pi^2}\mathbb{E}^{\pi^1, \pi^2}\left[\sum_{t=1}^\infty \alpha^{t-1}r_t|s_1 = s\right]$. It is known that for any discount factor $0 < \alpha < 1$, discounted SGs always have maximin stationary policies $\pi_\alpha = (\pi_\alpha^1, \pi_\alpha^2)$ which attain the game value $V_\alpha^*(s)$ for all $s$. We next show that the span of $V_\alpha^*$ is uniformly bounded by $D$ under Assumption 2.

**Lemma E.2.** *[14] Suppose given a stochastic game $M = (\mathcal{S}, \mathcal{A}, r, p)$, where $0 \le r(s, a^1, a^2) \le 1$. Suppose $\forall s, s' \in \mathcal{S}$ and for any $\pi^2 \in \Pi^{SR}$ for Player 2, there exists a $\pi^1 \in \Pi^{SR}$ for Player 1 such that the mean first passage time $T_{s,s'}^{\pi^1; \pi^2} \le D$. Then we have $|V_\alpha^*(s) - V_\alpha^*(s')| \le D$, $\forall s, s' \in \mathcal{S}$, for all $0 < \alpha < 1$.*

*Proof.* Fix $s, s' \in \mathcal{S}$. Fix a discount factor $0 < \alpha < 1$. For a fixed pair of maximin stationary policies $\pi_\alpha = (\pi_\alpha^1, \pi_\alpha^2) \in \Pi^{SR} \times \Pi^{SR}$, the discounted value function satisfies $V_\alpha^*(s) = r(s, \pi_\alpha^1, \pi_\alpha^2) + \alpha \sum_{s'} p(s'|s, \pi_\alpha^1, \pi_\alpha^2) V_\alpha^*(s')$. Since for any $\pi^1 \in \Pi^{SR}$, $V_\alpha^*(s) \geq r(s, \pi^1, \pi_\alpha^2) + \alpha \sum_{s'} p(s'|s, \pi^1, \pi_\alpha^2) V_\alpha^*(s')$, thus recursively, for any time step $T \geq 1$, we have

$$V_\alpha^*(s) \geq \mathbb{E}_s^{\pi^1, \pi_\alpha^2} \Big[ \sum_{t=1}^{T-1} \alpha^{t-1} r_t + \alpha^{T-1} V_\alpha^*(s_T) \Big],$$

where $\mathbb{E}_s^{\pi^1, \pi^2}[\cdot] = \mathbb{E}_s[\cdot \,|\, \pi^1, \pi^2]$ denote the expectation conditioned on initial state being $s$, and the players executing the policy pair $(\pi^1, \pi^2)$. Hence for any stopping time $\tau$,

$$V_\alpha^*(s) \geq \mathbb{E}_s^{\pi^1, \pi_\alpha^2} \Big[ \sum_{t=1}^{\tau-1} \alpha^{t-1} r_t + \alpha^{\tau-1} V_\alpha^*(s_\tau) \Big].$$

In particular, by choosing $\tau$ as the hitting time of $s'$ from $s$,

$$\begin{aligned}
V_\alpha^*(s) &\geq \mathbb{E}_s^{\pi^1, \pi_\alpha^2} \Big[ \sum_{t=1}^{\tau-1} \alpha^{t-1} r_t \Big] + \mathbb{E}_s \Big[ \alpha^\tau \Big| \pi^1, \pi_\alpha^2 \Big] V_\alpha^*(s') \\
&\geq \alpha^{\mathbb{E}_s[\tau | \pi^1, \pi_\alpha^2]} V_\alpha^*(s') = \alpha^{T_{s \to s'}^{\pi^1, \pi_\alpha^2}} V_\alpha^*(s') \\
&\geq V_\alpha^*(s') - (1 - \alpha)(T_{s \to s'}^{\pi^1, \pi_\alpha^2}) V_\alpha^*(s') \\
&\geq V_\alpha^*(s') - T_{s \to s'}^{\pi^1, \pi_\alpha^2} \\
&\geq V_\alpha^*(s') - D.
\end{aligned}$$

For the first inequality we used $V_\alpha(s_\tau) = V_\alpha(s')$ and for the second, the non-negativity of $r(s, a)$ and Jensen's inequality. The equality holds since the expected value of hitting time is the mean first passage time $T_{s \to s'}^{\pi^1, \pi_\alpha^2}$. The third inequality is essentially $\alpha^x \geq (\alpha - 1)x + 1$ for $x \geq 1$; the fourth $(1 - \alpha)V_\alpha \leq 1$. For the last inequality we used the assumption that there exists some $\pi^1$ for which $T_{s \to s'}^{\pi^1, \pi_\alpha^2} \leq D$. $\qquad\square$

**Lemma E.3.** *[12] Suppose $|V_\alpha^*(s) - V_\alpha^*(s')|$ is uniformly bounded for all $0 < \alpha < 1$ and for any $s, s' \in \mathcal{S}$. Then there exist a pair of maximin stationary policies $\pi = (\pi^{1*}, \pi^{2*})$ attaining the game value $\rho^*$ which is independent of the initial state and a bounded function $h(\cdot)$ for which the following equations hold. For all state $s$,*

$$\rho^* + h(s) = \max_{\pi^1 \in \Pi^{SR}} \Big\{ r(s, \pi^1, \pi^{2*}) + \sum_{s'} p(s'|s, \pi^1, \pi^{2*}) h(s') \Big\},$$

$$\rho^* + h(s) = \min_{\pi^2 \in \Pi^{SR}} \Big\{ r(s, \pi^{1*}, \pi^2) + \sum_{s'} p(s'|s, \pi^{1*}, \pi^2) h(s') \Big\}.$$

*Proof.* For any discount factor $0 < \alpha < 1$,

$$V_\alpha^*(s) = \max_{\pi^1} \{ r(s, \pi^1, \pi_\alpha^2) + \alpha \sum_{s'} p(s'|s, \pi^1, \pi_\alpha^2) V_\alpha^*(s') \},$$

$$V_\alpha^*(s) = \min_{\pi^2} \{ r(s, \pi_\alpha^1, \pi^2) + \alpha \sum_{s'} p(s'|s, \pi_\alpha^1, \pi^2) V_\alpha^*(s') \}.$$

Subtracting both sides by $V_\alpha^*(s_1)$ for some fixed state $s_1$, and defining $v_\alpha(s) := V_\alpha^*(s) - V_\alpha^*(s_1)$, we get, for all $s$,

$$v_\alpha(s) = \max_{\pi^1} \{ r(s, \pi^1, \pi_\alpha^2) - (1 - \alpha) V_\alpha^*(s_1) + \alpha \sum_{s'} p(s'|s, \pi^1, \pi_\alpha^2) v_\alpha(s') \},$$

$$v_\alpha(s) = \min_{\pi^2} \{ r(s, \pi_\alpha^1, \pi^2) - (1 - \alpha) V_\alpha^*(s_1) + \alpha \sum_{s'} p(s'|s, \pi_\alpha^1, \pi^2) v_\alpha(s') \}.$$

Since $-D \leq v_\alpha(s) \leq D$, $0 \leq (1 - \alpha) V_\alpha^*(s_1) \leq 1$ and $\pi_\alpha^i \in \Pi^{SR}$, $(i = 1, 2)$, all of which are contained in compact subsets/spaces, by using diagonalization argument and by Lebesgue

---
**Algorithm 2** Value Iteration with Schweitzer transform
---
**Input:** $M = (\mathcal{S}, \mathcal{A}^1 \times \mathcal{A}^2, r, p), 0 < \gamma < 1, 0 < \alpha < 1$.
**Initialization:** $v_0 \equiv 0$.
**repeat for** $i = 1, 2, ...$
$v_i = (1 - \alpha) \operatorname{val} \{r + P v_{i-1}\} + \alpha v_{i-1}$.
**until** $\operatorname{sp}(v_i - v_{i-1}) \leq (1 - \alpha)\gamma$.

---

convergence theorem, we can obtain a sequence $\alpha_k \to 1$, a bounded function $h$, and a constant $\rho^*$ such that $v_{\alpha_k}(\cdot) \to h(\cdot), (1 - \alpha_k) V_{\alpha_k}^*(s_1) \to \rho^*, \pi_{\alpha_k}^i \to \pi^{i*}, (i = 1, 2)$, and

$$\alpha_k \sum_{s'} p(s'|s, \pi^1, \pi_{\alpha_k}^2) v_{\alpha_k}(s') \to \sum_{s'} p(s'|s, \pi^1, \pi^{2*}) h(s'),$$

$$\alpha_k \sum_{s'} p(s'|s, \pi_{\alpha_k}^1, \pi^2) v_{\alpha_k}(s') \to \sum_{s'} p(s'|s, \pi^{1*}, \pi^2) h(s'),$$

as $k \to \infty$. Hence in the limit, for all state $s$,

$$\rho^* + h(s) = \max_{\pi^1 \in \Pi^{\text{SR}}} \left\{ r(s, \pi^1, \pi^{2*}) + \sum_{s'} p(s'|s, \pi^1, \pi^{2*}) h(s') \right\},$$

$$\rho^* + h(s) = \min_{\pi^2 \in \Pi^{\text{SR}}} \left\{ r(s, \pi^{1*}, \pi^2) + \sum_{s'} p(s'|s, \pi^{1*}, \pi^2) h(s') \right\}.$$

$\square$

# F  MAXIMIN-EVI and Its Convergence

As noted in Section 4.1, MAXIMIN-EVI proceeds simply by applying value iteration (Algorithm 2) on $M^+$. The output of the algorithm is a value vector with tolerable errors. The $\operatorname{val}\{r + P v_{i-1}\}$ term in Algorithm 2 becomes

$$\operatorname{val} \left\{ r(s, a^{1+}, a^2) + \sum_{s'} p^+(s'|s, a^{1+}, a^2) v_{i-1}(s')) \right\}$$

$$= \operatorname{val} \left\{ r(s, a^1, a^2) + \max_{\tilde{p}(\cdot) \in \mathcal{P}_k(s, a^1, a^2)} \sum_{s'} \tilde{p}(s') v_{i-1}(s')) \right\}. \tag{17}$$

The inner maximization can be efficiently solved with linear programming. The MAXIMIN-EVI$(\mathcal{M}_k, \gamma_k)$ in UCSG is then done by running Algorithm 2 with the evaluation of (17) in every iteration.

The following three lemmas characterize the convergence of the algorithm, and the properties of its outputs when converged. Lemma F.1 gaurantees that MAXIMIN-EVI converges. Lemma F.2 shows that when the algorithm halts, the output policy's worst-case average reward does not deviate from the maximin reward by more than $\gamma$. Lemma F.3 shows that the output value vector has a span no more than $D$.

**Lemma F.1** (Theorem 4 in [34]). *Suppose that Assumption 2 holds for some SG $M$. Then performing Value Iteration with Schweitzer transform on $M$ converges asymptotically.*

*Proof of Lemma F.1.* If Assumption 2 holds, then the Bellman equation holds with an initial-state independent game value by Theorem E.1. Then by Theorem 4 of [34], the value iteration with Schweitzer transform converges. $\square$

**Lemma F.2.** *Suppose that Assumption 2 holds for some stochastic game $M$. Let $\{v_i\}$ be the value sequence in the Value Iteration algorithm. Let $N$ be the index when iteration halts, i.e., $sp(v_{N+1} - v_N) \leq (1 - \alpha)\gamma$. Let $\pi^1 := \operatorname{solve}_1\{r + P v_N\}$. Then $\pi^1$ is $\gamma$-optimal in the sense that $\min_{\pi^2} \rho(M, \pi^1.\pi^2) \geq \rho^*(M) - \gamma$.*

*Proof of Lemma F.2.* Let $D = \min_s \{v_{N+1}(s) - v_N(s)\}$ and $U = \max_s \{v_{N+1}(s) - v_N(s)\}$. Then
$$D\mathbf{e} + v_N \leq v_{N+1} = (1-\alpha)\,\mathrm{val}\{r + Pv_N\} + \alpha v_N \leq (1-\alpha)(r_\pi + P_\pi v_N) + \alpha v_N,$$

where $\pi = (\pi^1, \pi^2)$ for any $\pi^2 \in \Pi^{\mathrm{SR}}$. Let $P_\pi^* = \lim_{T\to\infty} \frac{1}{T}\sum_{t=1}^T P_\pi^{t-1}$ be the Cesaro limit of $P_\pi$. Applying it on both sides of the inequality, we get $D\mathbf{e} \leq (1-\alpha)P_\pi^* r_\pi = (1-\alpha)\rho(M, \pi^1, \pi^2, \cdot)$, or $D \leq (1-\alpha)\rho(M, \pi^1, \pi^2, s)$, $\forall s, \pi^2$. Let $\pi^* = (\pi^{1*}, \pi^{2*})$ be the optimal policy pair and $\rho^*(M)$ be their maximin value, then $D \leq (1-\alpha)\rho(M, \pi^1, \pi^{2*}, s) \leq (1-\alpha)\rho^*(M)$. In a similar way, one can prove that $U \geq (1-\alpha)\rho^*(M)$. Since we assume $U - D \leq (1-\alpha)\gamma$, we have $D \geq (1-\alpha)(\rho^*(M) - \gamma)$. Therefore, $\pi^1$ is $\gamma$-optimal in the sense that $\forall \pi^2$, $\rho(M, \pi^1, \pi^2, s) \geq \rho^*(M) - \gamma$. $\qquad\square$

**Lemma F.3.** *If Assumption 2 holds for some model $M$, then value iteration procedure in Algorithm 2 will always produce value functions with spans bounded by $D$. That is,*
$$\mathrm{sp}\,(v_i) \leq D, \quad \forall i.$$

*Proof.* Note that value iteration with Schweitzer transform is equivalent to the following procedure. First modify the transition kernel and reward by $p_\alpha(s'|s, a^1, a^2) = (1-\alpha)p(s'|s, a^1, a^2) + \alpha\delta_{s,s'}$ and $r_\alpha(s, a^1, a^2) = (1-\alpha)r(s, a^1, a^2) + \alpha 0$; then do the normal value iteration by $v_i = \mathrm{val}\{r_\alpha + P_\alpha v_{i-1}\}$. By the principle of dynamic programming, $v_i$ is the maximin expected reward in the $i$-step game under the transformed model.

The transformed model is equivalent to the system where at each time step, the state remains same as the previous one with probability $\alpha$, and within that step there is no reward obtained/paid.

Clearly, in this new game, the advantage of starting from state $s$ than starting from state $s'$ (which can be calculated by $v_i(s) - v_i(s')$) is no more than that in the original game. In the original game, by a similar argument as Remark 8 in [19], this advantage difference is bounded by $D$. This then implies the argument in the lemma. $\qquad\square$

## G  Proof of Lemma 5.2

Lemma 5.2 directly follows from Lemma G.1 and G.2.

In this proof, we borrow the technique used in [22] and [9] to bound the number of steps with inaccurate transition probabilities (while they use this technique to bound the number of steps with inaccurate game value). Note here again that $\pi_t(\cdot)$ can represent any history-dependent policy, and we hide its parameter $H_t = (s_1, a_1, r_1, ..., s_t)$ inside the subscript of $t$.

Define the *importance* of a joint action $a$ at time $t$ as
$$\iota_t(a) := \max\left\{z_j : z_j \leq \frac{\pi_t(a)}{w_{\min}}\right\},$$
and the its *knownness* as
$$\kappa_t(a) := \max\left\{z_j : z_j \leq \frac{n_{k(t)}(s_t, a)}{m\pi_t(a)}\right\},$$
with $z_1 = 0$, $z_j = 2^{j-2}$ $\forall j = 2, 3, ...$, and some pre-defined $w_{\min} > 0$, $m > 0$. Note that we can always define them in hindsight even though the learner does not know $\pi_t^2$. These two amounts make partitions to the action set available at $s_t$. The partitioning is based on the actions' probability of being selected at time $t$ (i.e., $\pi_t(a)$), and the accuracy it has been estimated (the larger $n_{k(t)}(s_t, a)$, the more accurate). Intuitively, the larger $\kappa_t(a)$, the less likely will action $a$ contribute to inaccurate transition probability estimation. Define the partitions by $X_{t,\kappa,\iota} := \{a : \kappa_t(a) = \kappa \text{ and } \iota_t(a) = \iota\}$, $\forall \kappa, \iota$.

If we let $w_{\min} = \frac{\varepsilon}{3\sqrt{2\ln(1/\delta)}A}$ and $m = \frac{5\log_2^2(T/w_{\min})\ln(1/\delta)}{\varepsilon^2}$, with some $0 < \varepsilon < 1$, we can prove the following lemmas.

**Lemma G.1.** *For any $s$, any $\kappa$ and any $\iota > 0$, with probability at least $1 - \delta$,*
$$\sum_{t=1}^T \mathbb{1}_{s_t=s}\mathbb{1}_{|X_{t,\kappa,\iota}|>\kappa} = \mathcal{O}\left(\frac{A\log_2^2(T/w_{\min})\ln(1/\delta)}{\varepsilon^2}\right).$$

**Lemma G.2.** *If for all $\kappa$ and all $\iota > 0$ we have $|X_{t,\kappa,\iota}| \leq \kappa$, then for any plausible $\tilde{p}$ in the confidence set $\mathcal{M}_{k(t)}$, $|\tilde{p}(s'|s_t, \pi_t) - p(s'|s_t, \pi_t)| \leq \varepsilon$ for all $s'$.*

## G.1 Proof of Lemma G.1

We prove Lemma G.1 with the help of Lemma G.3 and G.4.

**Lemma G.3.** *For any $s, \kappa$, and $\iota > 0$, $\sum_{t=1}^{T} \mathbb{1}_{s_t=s} \mathbb{1}_{a_t \in X_{t,\kappa,\iota}} \leq 6Am(\kappa+1)\iota w_{\min}$.*

*Proof.* First fix $a$. By the definition of importance, if $a \in X_{t,\kappa,\iota}$, then $\iota w_{\min} \leq \pi_t(a) < 2\iota w_{\min}$. In the case $\kappa > 0$, we also have $m\kappa\pi_t(a) \leq n_{k(t)}(s_t, a) < 2m\kappa\pi_t(a)$. They two together imply $m\kappa\iota w_{\min} \leq n_{k(t)}(s_t, a) < 4m\kappa\iota w_{\min}$. This last inequality says that any $(s,a)$ cannot be sampled in the partition $(\kappa, \iota)$ for more than about $3m\kappa\iota w_{\min}$ times. This is because when $(s,a)$ is sampled once (i.e., $s_t = s, a_t = a$), $n_{k(t)}(s,a)$ will be increased by one, and this cannot happen for more than $4m\kappa\iota w_{\min} - m\kappa\iota w_{\min}$ times while $(s,a) \in X_{t,\kappa,\iota}$. Since UCSG only updates $n_k(s,a)$ when new phases start and doubling the sample count of a state-action triple incurs a phase change, we use a more conservative bound of $6m\kappa\iota w_{\min}$. That is, we have

$$\sum_{t=1}^{T} \mathbb{1}_{s_t=s} \mathbb{1}_{a_t=a} \mathbb{1}_{a \in X_{t,\kappa,\iota}} \leq 6m\kappa\iota w_{\min}. \tag{18}$$

In the case $\kappa = 0$, we have $n_{k(t)}(s_t, a) < m\pi_t(a) < 2m\iota w_{\min}$. Thus similarly, the sample counts of $(s,a)$ in the partition $(\kappa, \iota)$ cannot exceed $4m\iota w_{\min}$. The cases of $\kappa > 0$ and $\kappa = 0$ can then be combined into a single one:

$$\sum_{t=1}^{T} \mathbb{1}_{s_t=s} \mathbb{1}_{a_t=a} \mathbb{1}_{a \in X_{t,\kappa,\iota}} \leq 6m(\kappa+1)\iota w_{\min}. \tag{19}$$

Summing (19) over all actions leads to the statement in the lemma. $\square$

Now we sketch the argument of the next lemma. When $\iota > 0$, each action in $X_{t,\kappa,\iota}$ are to be sampled with probability no less than $\iota w_{\min}$. If furthermore $|X_{t,\kappa,\iota}|$ is large, the probability that some $a \in X_{t,\kappa,\iota}$ is sampled will be also large. However by Lemma G.3, the total times elements in partition $(\kappa, \iota)$ are sampled are upper bounded. Therefore, we can conclude that $|X_{t,\kappa,\iota}|$ cannot be large for too many steps. Formally, we have

**Lemma G.4.** *With probability at least $1 - \delta$,*

$$\sum_{t=1}^{T} \mathbb{1}_{s_t=s} \mathbb{1}_{a_t \in X_{t,\kappa,\iota}} \geq \frac{1}{2}(\kappa+1)\iota w_{\min} \sum_{t=1}^{T} \mathbb{1}_{s_t=s} \mathbb{1}_{|X_{t,\kappa,\iota}|>\kappa} - \frac{9}{2}\log(T\delta^{-1}).$$

*Proof.* To prove Lemma G.4, we need the help of Lemma B.2.

Let $\mathcal{F}_{t-1} = H_t = (s_1, a_1, r_1 \cdots, s_{t-1}, a_{t-1}, r_{t-1}, s_t)$ and

$$Y_t = q_t - \mathbb{1}_{s_t=s} \mathbb{1}_{|X_{t,\kappa,\iota}|>\kappa} \mathbb{1}_{a_t \in X_{t,\kappa,\iota}},$$

where we define

$$q_t := \mathbb{1}_{s_t=s} \mathbb{1}_{|X_{t,\kappa,\iota}|>\kappa} \Pr\left\{ a_t \in X_{t,\kappa,\iota} \middle| s_t = s, |X_{t,\kappa,\iota}| > \kappa \right\}.$$

Then Lemma B.2's conditions are met with $b = 1$. Moreover,

$$V_T = \sum_{t=1}^{T} q_t(1 - q_t) \leq \sum_{t=1}^{T} q_t.$$

Substituting them into Lemma B.2 and rearraging terms, we get that with probability $\geq 1 - \delta$,

$$\sum_{t=1}^{T} \mathbb{1}_{s_t=s} \mathbb{1}_{|X_{t,\kappa,\iota}|>\kappa} \mathbb{1}_{a_t \in X_{t,\kappa,\iota}} \geq \left(\sum_{t=1}^{T} q_t\right) - 2\sqrt{\left(\sum_{t=1}^{T} q_t\right)\log(T\delta^{-1})} - \sqrt{5}\log(T\delta^{-1}).$$

Solving the above inequality with respect to $\sqrt{\sum_{t=1}^{T} q_t}$, we can bound with probability $\geq 1 - \delta$ that

$$\sum_{t=1}^{T} \mathbb{1}_{s_t=s} \mathbb{1}_{|X_{t,\kappa,\iota}|>\kappa} \mathbb{1}_{a_t \in X_{t,\kappa,\iota}} \geq \frac{1}{2}\sum_{t=1}^{T} q_t - \frac{9}{2}\log(T\delta^{-1}). \tag{20}$$

Finally we look at $q_t$. Since each action in $X_{t,\kappa,\iota}$ are drawn at time $t$ with probability at least $\iota w_{\min}$, we have

$$q_t \geq \mathbb{1}_{s_t=s}\mathbb{1}_{|X_{t,\kappa,\iota}|>\kappa}\left(\sum_{a\in X_{t,\kappa,\iota}}\iota w_{\min}\right) \geq (\kappa+1)\iota w_{\min}\mathbb{1}_{s_t=s}\mathbb{1}_{|X_{t,\kappa,\iota}|>\kappa}. \tag{21}$$

Combining (20), (21), and noting that $\mathbb{1}_{s_t=s}\mathbb{1}_{a_t\in X_{t,\kappa,\iota}} \geq \mathbb{1}_{s_t=s}\mathbb{1}_{|X_{t,\kappa,\iota}|>\kappa}\mathbb{1}_{a_t\in X_{t,\kappa,\iota}}$ concludes the proof. $\square$

*Proof of Lemma G.1.* Combining Lemma G.3 and G.4, we have

$$\sum_{t=1}^{T}\mathbb{1}_{s_t=s}\mathbb{1}_{|X_{t,\kappa,\iota}|>\kappa} \leq 12Am + \frac{9}{(\kappa+1)\iota w_{\min}} \tag{22}$$

with probability no less than $1-\delta$. The lemma is then proved by substituting the selection of $m$ and $w_{\min}$ into (22), and using $\kappa+1 \geq 1$, $\iota \geq 1$. $\square$

### G.2 Proof of Lemma G.2

*Proof of Lemma G.2.*

$$|\tilde{p}(s'|s_t,\pi_t) - p(s'|s_t,\pi_t)| \leq \sum_a \pi_t(a)|\tilde{p}(s'|s_t,a) - p(s'|s_t,a)|$$

$$\leq \sqrt{2\ln\frac{1}{\delta_1}}\left(\sum_{a:\iota_t(a)=0}\sqrt{\frac{\pi_t(a)^2}{n_{k(t)}(s_t,a)}} + \sum_{\substack{\kappa,\iota:\\\iota>0}}\sum_{a\in X_{t,\kappa,\iota}}\sqrt{\frac{\pi_t(a)^2}{n_{k(t)}(s_t,a)}}\right)$$

$$\leq \sqrt{2\ln\frac{1}{\delta_1}}\left(Aw_{\min} + \sum_{\substack{\kappa,\iota:\\\iota>0}}\sqrt{|X_{t,\kappa,\iota}|\sum_{a\in X_{t,\kappa,\iota}}\frac{\pi_t(a)^2}{n_{k(t)}(s_t,a)}}\right)$$

$$\leq \sqrt{2\ln\frac{1}{\delta_1}}\left(Aw_{\min} + \sum_{\substack{\kappa,\iota:\\\iota>0,\kappa>0}}\sqrt{\kappa\sum_{a\in X_{t,\kappa,\iota}}\frac{\pi_t(a)}{m\kappa}}\right)$$

$$\leq \sqrt{2\ln\frac{1}{\delta_1}}\left(Aw_{\min} + \sqrt{|\mathcal{K}\times\mathcal{I}|\sum_{\substack{\kappa,\iota:\\\iota>0,\kappa>0}}\sum_{a\in X_{t,\kappa,\iota}}\frac{\pi_t(a)}{m}}\right)$$

$$\leq \sqrt{2\ln\frac{1}{\delta_1}}\left(Aw_{\min} + \sqrt{\frac{|\mathcal{K}\times\mathcal{I}|}{m}}\right),$$

where $\mathcal{K}$ and $\mathcal{I}$ are the set of effective $\kappa$'s and $\iota$'s in the above summation (only partitions with $\iota > 0$ and $\kappa > 0$ are relevant). By definition, there are at most $\log_2\left(\frac{1}{w_{\min}}\right)$ different values of $\iota$ for $\iota > 0$, and $\log_2\left(\frac{T}{mw_{\min}}\right) \leq \log_2\left(\frac{T}{w_{\min}}\right)$ different values for $\kappa > 0$ when $\iota > 0$. The second inequality is by the definition of the confidence set; the third and the fifth are by Cauchy's inequality; the fourth is by the assumption of the lemma. Substituting the values of $w_{\min}$ and $m$ into the last expression, we can get the desired result. $\square$

## H  Proofs of Lemma 5.1 and 5.3

To prove Lemma 5.1 and 5.3, the following lemma is a useful tool. In the following texts, we let $v_k(s) := \sum_{t=t_k}^{t_{k+1}-1}\mathbb{1}_{s_t=s}$, and write the joint policy $(\pi_k^1,\pi_k^2)$ as $\bar{\pi}_k$.

**Lemma H.1.** *Let $v \geq 1$. Then $\forall s$, with high probability, $\sum_k T_k\mathbb{1}_{v_k(s)\leq v} = \tilde{\mathcal{O}}(vDSA)$.*

*Proof.* Under Assumption 1, the times a state is visited within an interval of length $D$ is in average no less than 1 (no matter what policies the players play). Consider any arbitrarily chosen time frame $[\tau,\tau') \subset [1,T]$. In this time frame, there are $\lfloor\frac{\tau'-\tau}{2D}\rfloor$ intervals each with length $2D$. By Markov's

inequality, the probability $s$ is visited at least once within each interval is lower bounded by $\frac{1}{2}$. With Azuma-Hoeffding's inequality, we have with probability at least $1 - \frac{\delta}{T^2}$ that

$$
\begin{aligned}
\sum_{t=\tau}^{\tau'-1} \mathbb{1}_{s_t=s} &\geq \frac{1}{2}\left\lfloor \frac{\tau'-\tau}{2D} \right\rfloor - \sqrt{\left\lfloor \frac{\tau'-\tau}{2D} \right\rfloor \log\left(\frac{T^2}{\delta}\right)} \\
&\geq \frac{1}{4}\left\lfloor \frac{\tau'-\tau}{2D} \right\rfloor - 4\log\left(\frac{T^2}{\delta}\right) \\
&\geq \frac{1}{4}\frac{\tau'-\tau}{2D} - \frac{1}{4} - 4\log\left(\frac{T^2}{\delta}\right),
\end{aligned}
\tag{23}
$$

where the second inequality is easily verified by substracting RHS from LHS, and the third inequality is by the property of the floor function. Using an union bound over all possible $\tau$ and $\tau'$, we get that (23) holds for all $\tau, \tau'$ with probability at least $1 - \delta$.

Now apply (23) to all phases $k$ with $v_k(s) \leq v$, and sum all of them up. Then we get

$$
\sum_{k:v_k(s)\leq v} v_k(s) \geq \sum_{k:v_k(s)\leq v}\left(\frac{T_k}{8D} - \frac{1}{4} - 4\log\left(\frac{T^2}{\delta}\right)\right)
$$

or

$$
\sum_{k:v_k(s)\leq v} T_k \leq 8D \sum_{k:v_k(s)\leq v}\left(v_k(s) + \frac{1}{4} + 4\log(T^2/\delta)\right).
\tag{24}
$$

Since there are at most $SA\log_2 T$ phases, the RHS of (24) is further bounded by $\left(8vD + 2D + 32D\log(T^2/\delta)\right)SA\log_2 T$, which proves this lemma. $\qquad\square$

*Proof of Lemma 5.1.* $\bar{\pi}_k^2$ is not well-defined if and only if there is a $s$ such that $v_k(s) = \sum_{t=t_k}^{t_{k+1}-1} \mathbb{1}_{s_t=s} = 0$. The proof is done by simply applying Lemma H.1 with $v = 1$ together with a union bound over all states $s$. $\qquad\square$

We prove Lemma 5.3 by proving the following Lemma H.2 and H.3.

**Lemma H.2.**

$$
\sum_k T_k \mathbb{1}\{\bar{\pi}_k^2 \text{ is well-defined}\}\mathbb{1}\left\{\exists s, \mu(M, \bar{\pi}_k, s) > \frac{2v_k(s)}{T_k}\right\} \leq \tilde{\mathcal{O}}(D^3 S^4 A) \text{ with high probability.}
$$

**Lemma H.3.**

$$
\sum_k T_k \mathbb{1}\{\bar{\pi}_k^2 \text{ is well-defined}\}\mathbb{1}\left\{\operatorname{sp}(h(M_k^1, \bar{\pi}_k, \cdot)) > 2D\right\} \leq \tilde{\mathcal{O}}(D^3 S^5 A) \text{ with high probability.}
$$

## H.1 Proof of Lemma H.2

*Proof of Lemma H.2.* This lemma says, the stationary distribution of the irreducible Markov chain induced by $\pi_k^1$ and $\bar{\pi}_k^2$ won't exceed the empirical distribution too much in most steps. To prove Lemma H.2, we will compare three transition probabilities:

$$
\bar{p}_k(s'|s) := p(s'|s, \pi_k^1, \bar{\pi}_k^2) = \frac{\sum_{t=t_k}^{t_{k+1}-1} \mathbb{1}_{s_t=s}p(s'|s, \pi_k^1, \pi_t^2)}{\sum_{t=t_k}^{t_{k+1}-1} \mathbb{1}_{s_t=s}},
$$

$$
\hat{p}_k(s'|s) := \frac{\sum_{t=t_k}^{t_{k+1}-1} \mathbb{1}_{s_t=s}\mathbb{1}_{s_{t+1}=s'}}{\sum_{t=t_k}^{t_{k+1}-1} \mathbb{1}_{s_t=s}},
$$

$$
\tilde{p}_k(s'|s) := \frac{\sum_{t=t_k}^{t_{k+1}-2} \mathbb{1}_{s_t=s}\mathbb{1}_{s_{t+1}=s'} + \mathbb{1}_{s_{t_{k+1}-1}=s}\mathbb{1}_{s_{t_k}=s'}}{\sum_{t=t_k}^{t_{k+1}-1} \mathbb{1}_{s_t=s}},
$$

and use perturbation analysis to claim that when they are close enough, the stationary distributions they induce will also be close. Here, $\hat{p}_k$ is constructed by counting empirical transitions. $\tilde{p}_k$ is

only slightly modified from $\hat{p}_k$: the last term in the numerator changes from $\mathbb{1}_{s_{t_{k+1}-1}}\mathbb{1}_{s_{t_{k+1}}}$ to $\mathbb{1}_{s_{t_{k+1}-1}}\mathbb{1}_{s_{t_k}}$. Under the condition that $\bar{\pi}_k^2$ is well-defined, $\sum_{t=t_k}^{t_{k+1}-1}\mathbb{1}_{s_t=s} \neq 0 \ \forall s$, which means that $\tilde{p}_k$ has non-zero probability to reach any states from any states, hence inducing an irreducible Markov chain. $\bar{p}_k$ also induces an irreducible Markov chain by Assumption 1. We denote the stationary distributions corresponding to $\bar{p}_k$ and $\tilde{p}_k$ by $\bar{\mu}_k$ and $\tilde{\mu}_k$.

We will see that $\tilde{\mu}_k$ is exactly the same as the empirical distribution (i.e., $\tilde{\mu}_k(s) = \frac{v_k(s)}{T_k}$). By Theorem C.2, when two transition probabilities are close enough, their stationary distributions will also be close. We will argue that except for a constant amount of steps, $|\bar{p}_k(s'|s) - \hat{p}_k(s'|s)| \leq \frac{1}{2DS}$ and $|\hat{p}_k(s'|s) - \tilde{p}_k(s'|s)| \leq \frac{1}{2DS}$ hold for all $s, s'$. When they both hold, we can use Theorem C.2 with $\|E\|_\infty = \max_{s,s'} |\bar{p}_k(s'|s) - \tilde{p}_k(s'|s)| \leq \frac{1}{DS}$ and bound $|\bar{\mu}_k(s) - \tilde{\mu}_k(s)| \leq \frac{1}{2}\bar{\mu}_k(s)$. This will directly imply $\bar{\mu}_k(s) \leq 2\tilde{\mu}_k(s) = \frac{2v_k}{T_k}$.

From the discussion above, Lemma H.2 is proved as long as the three following lemmas (Lemma H.4, H.5, H.6) are proved. $\qquad\square$

**Lemma H.4.**

$$\sum_k T_k \mathbb{1}\{\bar{\pi}_k^2 \text{ is well-defined}\} \mathbb{1}\left\{\exists s, s', |\bar{p}_k(s'|s) - \hat{p}_k(s'|s)| > \frac{1}{2DS}\right\} \leq \tilde{\mathcal{O}}(D^3 S^4 A) \ \text{w.h.p.}$$

**Lemma H.5.**

$$\sum_k T_k \mathbb{1}\{\bar{\pi}_k^2 \text{ is well-defined}\} \mathbb{1}\left\{\exists s, s', |\hat{p}_k(s'|s) - \tilde{p}_k(s'|s)| > \frac{1}{2DS}\right\} \leq \tilde{\mathcal{O}}(D^2 S^3 A) \ \text{w.h.p.}$$

**Lemma H.6.**

$$\tilde{\mu}_k(s) = \frac{v_k(s)}{T_k}.$$

*Proof of Lemma H.4.* Fix $s$, $s'$, and $k$. Consider the martingale difference sequence defined by $Y_t := \mathbb{1}_{s_t=s}\left(p(s'|s, \pi_{k(t)}^1, \pi_t^2) - \mathbb{1}_{s_{t+1}=s'}\right)$, where $k(t)$ denotes the phase to which time step $t$ belongs. By Lemma B.2, for any $\tau \leq T + 1$, with probability at least $1 - 2\delta/T$,

$$\left|\sum_{t=t_k}^{\tau-1} Y_t\right| \leq 2\sqrt{V_{t_k,\tau}\log(T^2\delta^{-1})} + \sqrt{5}\log(T^2\delta^{-1}). \tag{25}$$

Here $V_{t_k,\tau} = \sum_{t=t_k}^{\tau-1} q_t(1-q_t) \leq \sum_{t=t_k}^{\tau-1} q_t \leq \sum_{t=t_k}^{\tau-1}\mathbb{1}_{s_t=s}$ where $q_t := \mathbb{1}_{s_t=s}p(s'|s, \pi_{k(t)}^1, \pi_t^2) \leq \mathbb{1}_{s_t=s}$. With an union bound, we have that (25) holds for all $\tau$ with probability at least $1 - 2\delta$. Now pick $\tau$ to be $t_{k+1}$, and thus $V_{t_k,t_{k+1}} \leq \sum_{t=t_k}^{t_{k+1}-1}\mathbb{1}_{s_t=s} = v_k(s)$. Then we have

$$|\bar{p}_k(s'|s) - \hat{p}_k(s'|s)| = \left|\frac{\sum_{t=t_k}^{t_{k+1}-1}\mathbb{1}_{s_t=s}(p(s'|s, \pi_k^1, \pi_t^2) - \mathbb{1}_{s_{t+1}=s'})}{v_k(s)}\right|$$

$$\leq 2\sqrt{\frac{\log(T^2\delta^{-1})}{v_k(s)}} + \frac{\sqrt{5}\log(T^2\delta^{-1})}{v_k(s)}$$

with probability at least $1 - 2\delta$. Another union bound over $s'$ lets the above inequality holds for all $s'$ with probability at least $1 - 2S\delta$.

We need about $v_k(s) \geq 25D^2S^2\log(T^2\delta^{-1})$ to make $|\bar{p}_k(s'|s) - \hat{p}_k(s'|s)| \leq \frac{1}{2DS} \ \forall s'$ in the above inequality. By Lemma H.1, we see that the number of steps not satisfying this condition is upper bounded by $\tilde{\mathcal{O}}(D^3S^3A)$. Another union bound over $s$ proves the lemma. $\qquad\square$

*Proof of Lemma H.5.* By the construction of $\tilde{p}_k$, $|\tilde{p}_k(s'|s) - \hat{p}_k(s'|s)| \leq \frac{1}{v_k(s)} \ \forall s'$. Again, we use Lemma H.1 and set the threshold $v = \tilde{\Theta}(2DS)$ to make $|\tilde{p}_k(s'|s) - \hat{p}_k(s'|s)| \leq \frac{1}{2DS} \ \forall s'$. By Lemma H.1, this will hold except for $\tilde{\mathcal{O}}(D^2S^2A)$ steps. An union bound over states leads to the $\tilde{\mathcal{O}}(D^2S^3A)$ bound. $\qquad\square$

*Proof of Lemma H.6.* We only need to check whether the equation $\tilde{\mu}_k(s') = \sum_s \tilde{\mu}_k(s)\tilde{p}_k(s'|s)$ holds for all $s, s'$. Indeed,

$$\sum_s \tilde{\mu}_k(s)\tilde{p}_k(s'|s) = \sum_s \frac{v_k(s)}{T_k} \frac{\sum_{t=t_k}^{t_{k+1}-2} \mathbb{1}_{s_t=s}\mathbb{1}_{s_{t+1}=s'} + \mathbb{1}_{s_{t_{k+1}-1}=s}\mathbb{1}_{s_{t_k}=s'}}{v_k(s)}$$

$$= \frac{\sum_{t=t_k}^{t_{k+1}-2} \mathbb{1}_{s_{t+1}=s'} + \mathbb{1}_{s_{t_k}=s'}}{T_k} = \tilde{\mu}_k(s').$$

$\square$

## H.2 Proof of Lemma H.3

*Proof of Lemma H.3.* By Assumption 1, the maximum mean first passage time under model $M$ and policy pair $(\pi_k^1, \bar{\pi}_k^2)$ does not exceed $D$, i.e., $T^{\pi_k^1, \bar{\pi}_k^2}(M) \leq D$. Then by Theorem C.9, we know that if all transition probabilities in the Markov chain induced by $(M_k^1, \pi_k^1, \bar{\pi}_k^2)$ is perturbed from that induced by $(M, \pi_k^1, \bar{\pi}_k^2)$ within the amount of $\frac{1}{8DS^2}$, the former's maximum mean first passage time can be bounded by two times the latter's, i.e., $T^{\pi_k^1, \bar{\pi}_k^2}(M_k^1) \leq 2T^{\pi_k^1, \bar{\pi}_k^2}(M)$. This also implies that $(M_k^1, \pi_k^1, \bar{\pi}_k^2)$ induces an irreducible Markov chain. Finally, by Remark M.1, we have $\mathrm{sp}(h(M_k^1, \pi_k^1, \bar{\pi}_k^2, \cdot)) \leq T^{\pi_k^1, \bar{\pi}_k^2}(M_k^1)$. Combining the three inequalities above, we can have $\mathrm{sp}(h(M_k^1, \pi_k^1, \bar{\pi}_k^2, \cdot)) \leq 2D$. As a result, to prove this theorem, we only need to bound the number of steps in phases where there exist $s, s'$ such that the transition probability difference $|p_k^1(s'|s, \pi_k^1, \bar{\pi}_k^2) - p(s'|s, \pi_k^1, \bar{\pi}_k^2)|$ is larger than $\frac{1}{8DS^2}$ ($p_k^1$ is the transition kernel of $M_k^1$). We define the event $E_k(s) = \{\exists s', |p_k^1(s'|s, \pi_k^1, \bar{\pi}_k^2) - p(s'|s, \pi_k^1, \bar{\pi}_k^2)| > \frac{1}{8DS^2}\}$, and $E_k = \{\exists s, E_k(s) = 1\}$. Our goal is to prove $\sum_k T_k \mathbb{1}_{E_k} \leq \tilde{\mathcal{O}}(D^3 S^5 A)$.

Fix $k$. Suppose that $\bar{\pi}_k^2$ is well-defined. By the definition of $\bar{\pi}_k^2$ and the triangle inequality, we have

$$|p_k^1(s'|s, \pi_k^1, \bar{\pi}_k^2) - p(s'|s, \pi_k^1, \bar{\pi}_k^2)| \leq \frac{1}{v_k(s)} \sum_{t=t_k}^{t_{k+1}-1} \mathbb{1}_{s_t=s}|p_k^1(s'|s_t, \pi_t) - p(s'|s_t, \pi_t)|. \tag{26}$$

Define $\varepsilon_i := 2^{-i}$, and define

$$G_k(s, \varepsilon) = \{t \in [t_k, t_{k+1}) : s_t = s \text{ and } \varepsilon < \max_{s'}|p_k^1(s'|s_t, \pi_t) - p(s'|s_t, \pi_t)| \leq 2\varepsilon\},$$

and $g_k(s, \varepsilon) := |G_k(s, \varepsilon)|$, i.e., $g_k(s, \varepsilon)$ is the number of steps $t$ in phase $k$ such that $s_t = s$ and the maximum transition probability error at that step is between $\varepsilon$ and $2\varepsilon$. With these definitions, we can continue to upper bound (26) by

$$|p_k^1(s'|s, \pi_k^1, \bar{\pi}_k^2) - p(s'|s, \pi_k^1, \bar{\pi}_k^2)| \leq \frac{1}{v_k(s)} \sum_{t=t_k}^{t_{k+1}-1} \mathbb{1}_{s_t=s}|p_k^1(s'|s_t, \pi_t) - p(s'|s_t, \pi_t)|$$

$$\leq \frac{1}{v_k(s)} \left( \sum_{2\varepsilon_i > \frac{1}{24DS^2}} 2\varepsilon_i g_k(s, \varepsilon_i) + \frac{1}{24DS^2} v_k(s) \right)$$

$$= \frac{1}{24DS^2} + \sum_{i=1}^{\lfloor \log_2(48DS^2) \rfloor} \frac{2\varepsilon_i g_k(s, \varepsilon_i)}{v_k(s)}. \tag{27}$$

If $|p_k^1(s'|s, \pi_k^1, \bar{\pi}_k^2) - p(s'|s, \pi_k^1, \bar{\pi}_k^2)| > \frac{1}{8DS^2}$, then by (27) we have

$$\frac{v_k(s)}{24DS^2} \leq \sum_{i=1}^{\lfloor \log_2(48DS^2) \rfloor} \varepsilon_i g_k(s, \varepsilon_i). \tag{28}$$

Note that since steps counted in $G_k(s, \varepsilon)$ have maximum transition errors greater that $\varepsilon$, by Lemma 5.2, with high probability, $\sum_k g_k(s, \varepsilon)$ won't exceed $\frac{c_1 A}{\varepsilon^2}$, for some $c_1$ hides logarithmic terms. Now

sum the above equation over phases where $E_k(s)$ holds, we get that

$$\sum_{k:E_k(s)} \frac{v_k(s)}{24DS^2} \leq \sum_{k:E_k(s)} \sum_{i=1}^{\lfloor \log_2(48DS^2) \rfloor} \varepsilon_i g_k(s, \varepsilon_i) \leq \sum_{i=1}^{\lfloor \log_2(48DS^2) \rfloor} \frac{c_1 A}{\varepsilon_i} \leq 48c_1 DS^2 A \log_2(48DS^2)$$

or $\sum_{k:E_k(s)} v_k(s) \leq \tilde{\mathcal{O}}(D^2 S^4 A)$ holds with high probability. Similar to the proof of Lemma H.1, we use (23) and lower bound $\sum_{k:E_k(s)} v_k(s) \geq \sum_{k:E_k(s)} \left( \frac{T_k}{8D} - \tilde{\mathcal{O}}(1) \right)$. Combining the lower bound and the upper bound, we get $\sum_{k:E_k(s)} T_k \leq \tilde{\mathcal{O}}(D^3 S^4 A)$. Finally, summing over $s$, we get the desired bound. □

## I   Proofs of Lemma 5.4 and 5.5

*Proof of Lemma 5.4.* Define notations: $\bar{\pi}_k = (\pi_k^1, \bar{\pi}_k^2)$, $\bar{p}_k(s'|s) := p(s'|s, \bar{\pi}_k)$, $\bar{h}_k(s) := h(M, \bar{\pi}_k, s)$, $\bar{\rho}_k := \rho(M, \bar{\pi}_k)$, $\bar{r}_k(s) := r(s, \bar{\pi}_k)$, $r_t := r(s_t, a_t)$.

By the construction of $\bar{\pi}_k^2$, we have

$$\bar{p}_k(s'|s) = \sum_{a^2} \frac{\sum_{t=t_k}^{t_{k+1}-1} \mathbb{1}_{s_t=s} \pi_t(a^2)}{v_k(s)} p(s'|s, \pi_k^1, a^2) = \frac{1}{v_k(s)} \sum_{t=t_k}^{t_{k+1}-1} \mathbb{1}_{s_t=s} p(s'|s, \pi_k^1, \pi_t^2) \quad (29)$$

and

$$\bar{r}_k(s) = \sum_{a^2} \frac{\sum_{t=t_k}^{t_{k+1}-1} \mathbb{1}_{s_t=s} \pi_t^2(a^2)}{v_k(s)} r(s, \pi_k^1, a^2) = \frac{1}{v_k(s)} \sum_{t=t_k}^{t_{k+1}-1} \mathbb{1}_{s_t=s} r(s, \pi_k^1, \pi_t^2). \quad (30)$$

Our target in phase $k$ can be decomposed as:

$$T_k \bar{\rho}_k - \sum_{t=t_k}^{t_{k+1}-1} r_t = \sum_{t=t_k}^{t_{k+1}-1} (\bar{\rho}_k - \bar{r}_k(s_t)) + \sum_{t=t_k}^{t_{k+1}-1} (\bar{r}_k(s_t) - r_t), \quad (31)$$

Now manipulate individual terms.

$$\sum_{t=t_k}^{t_{k+1}-1} (\bar{\rho}_k - \bar{r}_k(s_t)) = \sum_{t=t_k}^{t_{k+1}-1} \left( \sum_{s'} \bar{p}_k(s'|s_t) \bar{h}_k(s') - \bar{h}_k(s_t) \right)$$

$$= \sum_{s,s'} v_k(s) \bar{p}_k(s'|s) \bar{h}_k(s') - \sum_{t=t_k}^{t_{k+1}-1} \bar{h}_k(s_t)$$

$$= \sum_{s,s'} \sum_{t=t_k}^{t_{k+1}-1} \mathbb{1}_{s_t=s} p(s'|s, \pi_k^1, \pi_t^2) \bar{h}_k(s') - \sum_{t=t_k}^{t_{k+1}-1} \bar{h}_k(s_t)$$

$$= \sum_{t=t_k}^{t_{k+1}-1} \sum_{s'} p(s'|s_t, \pi_k^1, \pi_t^2) \bar{h}_k(s') - \sum_{t=t_k}^{t_{k+1}-1} \bar{h}_k(s_t), \quad (32)$$

where the third equality follows from (29);

$$\sum_{t=t_k}^{t_{k+1}-1} (\bar{r}_k(s_t) - r_t) = \sum_s v_k(s) \bar{r}_k(s) - \sum_{t=t_k}^{t_{k+1}-1} r_t$$

$$= \sum_s \sum_{t=t_k}^{t_{k+1}-1} \mathbb{1}_{s_t=s} r(s, \pi_k^1, \pi_t^2) - \sum_{t=t_k}^{t_{k+1}-1} r_t$$

$$= \sum_{t=t_k}^{t_{k+1}-1} r(s_t, \pi_k^1, \pi_t^2) - \sum_{t=t_k}^{t_{k+1}-1} r_t, \quad (33)$$

where the second equality follows from (30). Substituting (32) and (33) into (31), we get

$$T_k \bar{\rho}_k - \sum_{t=t_k}^{t_{k+1}-1} r_t = \bar{h}_k(s_{t_{k+1}}) - \bar{h}_k(s_{t_k}) + \sum_{t=t_k}^{t_{k+1}-1} \left( Y_t^1 + Y_t^2 \right), \tag{34}$$

where $Y_t^1 := \left( \sum_{s'} p(s'|s_t, \pi_k^1, \pi_t^2)\bar{h}_k(s') - \bar{h}_k(s_{t+1}) \right)$, and $Y_t^2 := \left( r(s_t, \pi_k^1, \pi_t^2) - r_t \right)$. It seems that $Y_t^1$ and $Y_t^2$ have expectations of zero and should be able to be bounded with Bernstein's inequality. Nevertheless, one needs to be careful about that $\bar{h}_k$ depends on $\bar{\pi}_k^2$, which is only known after phase $k$ ends. In other words, $\bar{h}_k$ is not $\mathcal{F}_t$-measurable for $t \in \mathrm{ph}(k)$, where $\mathcal{F}_{t-1} := \{s_1, a_1, \cdots, s_t\}$. The solution is as follows. Let $\mathcal{D}$ be the set where $\bar{h}_k$ possibly lies. We discretize $\mathcal{D}$ and use the Bernstein bound on all discretization points. Finally, we use the fact that $\bar{h}_k$ is not far from the nearest discretization point to bound the sum of $Y_t^1$.

Let $\mathcal{D} := [-D, D]^S$, and thus $\bar{h}_k \in \mathcal{D}$. Clearly, there is a discretization $\mathcal{D}_d$ with $|\mathcal{D}_d| \leq (2DST)^S$ such that any $h \in \mathcal{D}$ can find some $h_d \in \mathcal{D}_d$ with $|h(s) - h_d(s)| \leq \frac{1}{ST} \forall s$. Now let $Y_t^{1(j)} := \left( \sum_{s'} p(s'|s_t, \pi_k^1, \pi_t^2)h^{(j)}(s') - h^{(j)}(s_{t+1}) \right)$ for every $h^{(j)} \in \mathcal{D}_d, j = 1, ..., (2DST)^S$. Now $Y_t^{1(j)}$'s are martingale difference sequences with respect to $\mathcal{F}_{t-1}$, so we can apply Azuma-Hoeffding's inequality and bound

$$\sum_k \sum_{t=t_k}^{t_{k+1}-1} Y_t^{1(j)} \leq \sqrt{\frac{\log((2DST)^S \delta^{-1})}{2} T(2D)^2} \tag{35}$$

with probability at least $1 - \frac{\delta}{(2DST)^S}$. Using the union bound, (35) holds for all $j$ with probability at least $1 - \delta$. Also, there exists a $j$ such that $\sum_{t=t_k}^{t_{k+1}-1} \left( Y_t^1 - Y_t^{1(j)} \right) \leq T_k \times \frac{2S}{ST} = \frac{2T_k}{T}$. Thus we have

$$\sum_k \sum_{t=t_k}^{t_{k+1}-1} Y_t^1 \leq \tilde{\mathcal{O}}(D\sqrt{ST}) \tag{36}$$

with high probability. We also have $\sum_k \sum_{t=t_k}^{t_{k+1}-1} Y_t^2 \leq \tilde{\mathcal{O}}(\sqrt{T})$ by Azuma-Hoeffding's inequality. Also, $\bar{h}_k(s_{t_{k+1}}) - \bar{h}_k(s_{t_k}) \leq 2D$. Collecting terms, we get the desired bound. $\square$

*Proof of Lemma 5.5.* First fix $k$. Denote the transition probabilities of the optimistically selected model $M_k^1$ by $p_k^1(\cdot|\cdot, \cdot, \cdot)$. In this proof, we define $\tilde{h}(\cdot) := h(M_k^1, \bar{\pi}_k, \cdot)$, $h(\cdot) := h(M, \bar{\pi}_k, \cdot)$, $\tilde{\mu}(\cdot) := \mu(M_k^1, \bar{\pi}_k, \cdot)$, $\mu(\cdot) := \mu(M, \bar{\pi}_k, \cdot)$, $\tilde{\rho} := \rho(M_k^1, \bar{\pi}_k)$, $\rho := \rho(M, \bar{\pi}_k)$, $r(\cdot) := r(s, \bar{\pi}_k)$, $\tilde{p}(s'|s) := p_k^1(s'|s, \bar{\pi}_k), p(s'|s) := p(s'|s, \bar{\pi}_k)$.

By Bellman equation and the properties of irreducible Markov chains, we have

$$\rho = r(s) + \sum_{s'} p(s'|s)h(s') - h(s)$$

$$\tilde{\rho} = r(s) + \sum_{s'} \tilde{p}(s'|s)\tilde{h}(s') - \tilde{h}(s)$$

for all $s$. Therefore, we can write (for any $s$)

$$\tilde{\rho} - \rho = \sum_{s'} \left( \tilde{p}(s'|s)\tilde{h}(s') - p(s'|s)h(s') \right) - \tilde{h}(s) + h(s)$$

$$= \sum_{s'} \left( \tilde{p}(s'|s) - p(s'|s) \right) \tilde{h}(s') + \sum_{s'} \left( p(s'|s) - \delta_{s,s'} \right) \left( \tilde{h}(s') - h(s') \right). \tag{37}$$

Thus,

$$T_k(\tilde{\rho} - \rho) = \sum_s T_k \mu(s)(\tilde{\rho} - \rho)$$

$$= \sum_s T_k \mu(s) \sum_{s'} \left( \tilde{p}(s'|s) - p(s'|s) \right) \tilde{h}(s')$$

$$\leq \sum_s T_k \mu(s) \|\tilde{p}(\cdot|s) - p(\cdot|s)\|_1 \operatorname{sp}(\tilde{h}), \tag{38}$$

where the second equality is by using (37) and the property of stationary distribution: $\sum_s \mu(s)(p(s'|s) - \delta_{s,s'}) = 0$. By the definition of $\tilde{p}$ and $p$, we have

$$\|\tilde{p}(\cdot|s) - p(\cdot|s)\|_1 \leq \sum_a \pi_t(a) \frac{\sum_{t=t_k}^{t_{k+1}-1} \mathbb{1}_{s_t=s} \|\tilde{p}(\cdot|s,a) - p(\cdot|s,a)\|_1}{v_k(s)}$$

$$= \frac{\sum_{t=t_k}^{t_{k+1}-1} \mathbb{1}_{s_t=s} \|\tilde{p}(\cdot|s,a_t) - p(\cdot|s,a_t)\|_1 + \sum_{t=t_k}^{t_{k+1}-1} Y_t}{v_k(s)} \tag{39}$$

where $Y_t := \mathbb{E}[q_t] - q_t$, and $q_t := \mathbb{1}_{s_t=s} \|\tilde{p}(\cdot|s,a_t) - p(\cdot|s,a_t)\|_1$. To apply Lemma B.2, we note that $|q_t| \leq 2$ and $V_T := \sum_{t=t_k}^{t_{k+1}-1} \mathbb{E}[Y_t^2|\mathcal{F}_{t-1}] \leq 2 \sum_{t=t_k}^{t_{k+1}-1} \mathbb{E}[q_t]$. Then we can bound

$$\sum_{t=t_k}^{t_{k+1}-1} (\mathbb{E}[q_t] - q_t) \leq 2\sqrt{\left(2 \sum_{t=t_k}^{t_{k+1}-1} \mathbb{E}[q_t]\right) \log(T^2\delta^{-1})} + 2\sqrt{5}\log(T^2\delta^{-1}) \tag{40}$$

with probability at least $1 - \delta$. (40) implies

$$\sum_{t=t_k}^{t_{k+1}-1} (\mathbb{E}[q_t] - q_t) \leq \sum_{t=t_k}^{t_{k+1}-1} q_t + 17\log(T^2\delta^{-1}). \tag{41}$$

Continuing (38) with the help of (39) and (41), we get

$$T_k(\tilde{\rho} - \rho) \leq 2D \sum_s T_k \mu(s) \frac{2\sum_{t=t_k}^{t_{k+1}-1} q_t + 17\log(T^2\delta^{-1})}{v_k(s)}$$

$$\leq 3D \sum_s \left(2 \sum_{t=t_k}^{t_{k+1}-1} q_t + 17\log(T^2\delta^{-1})\right)$$

$$\leq 6D \sum_{t=t_k}^{t_{k+1}-1} \|\tilde{p}(s_t,a_t) - p(s_t,a_t)\|_1 + \tilde{\mathcal{O}}(DS)$$

$$\leq 12D\sqrt{2S\ln\frac{1}{\delta_1}} \sum_{s,a} \frac{v_k(s,a)}{\sqrt{n_k(s,a)}} + \tilde{\mathcal{O}}(DS),$$

where we have used the assumptions in this lemma. Now sum over benign phases, we get

$$\sum_{k:\text{benign}} T_k \left(\rho(M_k^1, \bar{\pi}_k) - \rho(M, \bar{\pi}_k)\right) \leq \sum_k \sum_{s,a} \frac{v_k(s,a)}{\sqrt{n_k(s,a)}} \tilde{\mathcal{O}}(D\sqrt{S}) + \sum_k \tilde{\mathcal{O}}(DS) \tag{42}$$

$$\leq 2.5\sqrt{SAT}\tilde{\mathcal{O}}(D\sqrt{S}) + \tilde{\mathcal{O}}(DS^2A)$$

$$= \tilde{\mathcal{O}}(DS\sqrt{AT} + DS^2A).$$

with high probability. The last inequality is by the following Lemma together with Cauchy's inequality. $\qquad\square$

**Lemma I.1** (cf. Lemma 19 of [19]). *For any sequence $\{z_i\}, i = 1, ..., N$ with $0 \leq z_i \leq Z_{i-1} :=$ $\max\{1, \sum_{\ell=1}^{i-1} z_\ell\}$. Let $K$ be a subset of $\{1, ..., N\}$. Then we have*

$$\sum_{i \in K} \frac{z_i}{\sqrt{Z_{i-1}}} \leq (\sqrt{2} + 1)\sqrt{L},$$

*where $L := \sum_{i \in K} z_i$.*

*Proof.*

$$\sum_{i \in K} \frac{z_i}{\sqrt{Z_{i-1}}} \leq (\sqrt{2} + 1) \sum_{i \in K} \frac{z_i}{\sqrt{Z_i} + \sqrt{Z_{i-1}}}$$

$$= (\sqrt{2}+1) \sum_{i \in K} \frac{z_i \cdot (\sqrt{Z_i} - \sqrt{Z_{i-1}})}{(\sqrt{Z_i} + \sqrt{Z_{i-1}}) \cdot (\sqrt{Z_i} - \sqrt{Z_{i-1}})}$$

$$= (\sqrt{2}+1) \sum_{i \in K} (\sqrt{Z_i} - \sqrt{Z_{i-1}})$$

$$\leq (\sqrt{2}+1) \sum_{i \in K} (\sqrt{L_i} - \sqrt{L_{i-1}}) \leq (\sqrt{2}+1)\sqrt{L},$$

where $L_i := \sum_{\ell \in K : \ell \leq i} z_i$. We used the inequality

$$\sqrt{Z_i} - \sqrt{Z_{i-1}} \leq \sqrt{L_i} - \sqrt{L_{i-1}} \Leftrightarrow \frac{z_i}{\sqrt{L_i} + \sqrt{L_{i-1}}} \leq \frac{z_i}{\sqrt{Z_i} + \sqrt{Z_{i-1}}}.$$

$\square$

## J  Proofs of Lemma 6.1 and 6.2

*Proof of lemma 6.1.* Note that for any phase $k$ and any episode $i$ that fully lies in phase $k$, we have $\mathbb{E}\left[\sum_{t=\tau_i}^{\tau_{i+1}-1} r(s_t, a_t)\right] = V_H(M, \pi_k^1, \pi_i^2, s_{\tau_i})$. Therefore, the terms in $\sum_k \Delta_k^{(5)}$ form a martingale difference sequence with no more than $T/H$ terms. Furthermore, $0 \leq \sum_{t=\tau_i}^{\tau_{i+1}-1} r(s_t, a_t) \leq H$. By Lemma B.1, with probability $1-\delta$, we have $\sum_k \Delta_k^{(5)} \leq \sqrt{\frac{\log(\delta^{-1})}{2} \frac{T}{H} H^2} = \tilde{\mathcal{O}}(\sqrt{HT})$.  $\square$

*Proof of Lemma 6.2.* Suppose that the value iteration halts at iteration $N$, then under Assumption 2 and by the proof of Lemma F.2, we have

$$(1-\alpha)(\rho^*(M^+) - \gamma)\mathbf{e} \leq D\mathbf{e} \leq v_{N+1} - v_N = (1-\alpha)(\text{val}\{r + Pv_N\} - v_N). \quad (43)$$

Since $(M_k^1, p_k^1)$ is selected based on the $v_N$ when the value iteration halts, (43) is equivalent to

$$\rho^*(M^+) - \gamma \leq \min_{\pi^2} \left\{ r(s, \pi_k^1, \pi^2) + \sum_{s'} p_k^1(s'|s, \pi_k^1, \pi^2) v_N(s') \right\} - v_N(s). \quad (44)$$

Besides, the span of the vector $v_N$ is bounded by $D$ by Lemma F.3. Now we fix Player 1's policy as $\pi_k^1$ in the extended game, and let Player 2 run an $H$-step SG. The least amount Player 2 has to pay Player 1 in this SG is $\min_{\pi^2} V_H(M_k^1, \pi_k^1, \pi^2, s)$ (assuming that the game starts from $s$), which can be calculated by dynamic programming. The dynamic programming goes as follows: for $i = 0, ..., H-1$, for all $s$,

$$u_0(s) = 0,$$

$$u_{i+1}(s) = \min_a \{ r(s, \pi_k^1, a) + \sum_{s'} p_k^1(s'|s, \pi_k^1, a) u_i(s') \},$$

which, in its vector form, can be written as $u_{i+1} = \min_a \{ r_a + P_a u_i \}$ by denoting $r_a(\cdot) := r(\cdot, \pi_k^1, a)$ and $(P_a)_{ij} := p_k^1(j|i, \pi_k^1, a)$. We can re-write the induction procedure as

$$u_{i+1} - v_N = \min_a \{ r_a + P_a v_N + P_a (u_i - v_N) \} - v_N$$

without affecting the solution. By the property $\min\{u+v\} \geq \min\{u\} + \min\{v\}$, we have

$$u_{i+1} - v_N \geq \min_a \{ r_a + P_a v_N \} + \min_a \{ P_a (u_i - v_N) \} - v_N \quad (45)$$

By (44), $\min_a \{ r_a + P_a v_N \} - v_N \geq \rho^*(M^+) - \gamma$, and since $P_a$ is stochastic, $P_a(u_i - v_N) \geq \min_{s'} \{ u_i(s') - v_N(s') \}$. Combining them with (45), we have $u_{i+1}(s) - v_N(s) \geq \rho^*(M^+) - \gamma + \min_{s'} \{ u_i(s') - v_N(s') \}$ for all $s$. Then by induction, we can easily prove $u_i(s) - v_N(s) \geq i(\rho^*(M^+) - \gamma) + \min_{s'} \{ u_0(s') - v_N(s') \}$, and therefore, $u_i(s) \geq i(\rho^*(M^+) - \gamma) + v_N(s) - \max_{s'} v_N(s') \geq i(\rho^*(M^+) - \gamma) - D$.

Let $i = H$ and note that $\rho^*(M^+) = \max_{\tilde{M}} \max_{\pi^1} \min_{\pi^2} \rho(\tilde{M}, \pi^1, \pi^2) \geq \min_{\pi^2} \rho(M_k^1, \pi_k^1, \pi^2, s)$. The above result translates to $\min_{\pi^2} V_H(M_k^1, \pi_k^1, \pi^2, s) \geq H \min_{\pi^2} \rho(M_k^1, \pi_k^1, \pi^2, s) - D - H\gamma$, which bounds $\Delta_k^{(2)}$ by $\sum_{i \in \text{ph}(k)} (D + H\gamma)$.  $\square$

# K  Proof of Theorem K.1 and Lemma 6.3

**Theorem K.1.** *(Sample Complexity Bound of* UCSG. *cf.  Theorem 1 [9]) Given $\delta > 0$, with probability at least $1 - \delta$, for any $0 < \varepsilon < 1$,* UCSG *produces a sequence of policies $\pi_k^1$, that yield at most $\tilde{\mathcal{O}}\left(\frac{H^2 S^2 A}{\varepsilon^2}\right)$ episodes $i$ such that $|V_H(M, \pi_k^1, \pi_i^2, s_{\tau_i}) - V_H(M_k^1, \pi_k^1, \pi_i^2, s_{\tau_i})| > \varepsilon$.*

Theorem K.1 mainly follows from the following Lemma K.6 and K.7. In [9] the analysis of sample complexity is facilitated by partitioning the state-action space. The state-action pairs are grouped into different categories according to two indices. The first index, *importance*, measures in log-scale the relative occurrence frequency of $(s, a)$ with respect to a fixed constant under the policy. The second index, *knownness*, measures also in log-scale the ratio of the total number of observations to the occurrence frequency. Here we modify the the definition of weight, importance, and knownness for a state-joint action $(s, a) = (s, a^1, a^2)$ defined below to have a partition of the state-joint-action space $\mathcal{S} \times \mathcal{A} = \mathcal{S} \times \mathcal{A}^1 \times \mathcal{A}^2$ for each episode.

**Definition K.2.** *Define the weight of a state-joint-action pair $(s, a)$ under joint policy $\pi_i$ in episode $i$ as the expected occurrence frequency of $(s, a)$ in episode $i$,*

$$w_i(s, a) := \sum_{t=\tau_i}^{\tau_{i+1}-1} \mathbb{P}(s_t = s, a_t = a | a_t \sim \pi_i, s_{\tau_i}).$$

The setting in [9] is somewhat different from two-player zero-sum SGs. In the episodic RL setting after an episode is over, a new episode starts afresh with the same initial distribution $p_0$, while in the non-episodic setting, initial state $s_{\tau_i}$ in each episode is sampled from a different distribution. Initial state distributions do not matter that much in our setting except we need the initial state $s_{\tau_i}$ to compute the expected frequency $w_i(s, a)$.

**Definition K.3.** *Define the importance of a state-joint-action pair $(s, a)$ in episode $i$ as*

$$\iota_i(s, a) := \max\left\{z_j : z_j \leq \frac{w_i(s, a)}{w_{min}}\right\},$$

*where $z_1 = 0$ and $z_j = 2^{j-2} \; \forall j = 2, 3, ...$*

**Definition K.4.** *Define the knownness of a a state-joint-action pair $(s, a)$ in episode $i$ as*

$$\kappa_i(s, a) := \max\left\{z_j : z_j \leq \frac{n_{k(i)}(s, a)}{m w_i(s, a)}\right\},$$

where $z_1 = 0$ and $z_j = 2^{j-2} \; \forall j = 2, 3, ...$

**Definition K.5.** *We can now categorize state-joint-action pairs $(s, a)$ into subsets*

$$X_{i,\kappa,\iota} := \{(s, a) \in X_i : \kappa_i(s, a) = \kappa, \iota_i(s, a) = \iota\},$$

$$\text{and } \bar{X}_i = \mathcal{S} \times \mathcal{A} \backslash X_i, \text{ where } X_i = \{(s, a) \in \mathcal{S} \times \mathcal{A} : \iota_i(s, a) > 0\}.$$

In contrast to the original definitions [9] which are designated for each phase $k$ in the episodic RL setting, in our setting, weight $w_i(s, a)$, importance $\iota_i(s, a)$, knownness $\kappa_i(s, a)$ are now indexed for each episode $i$ because Player 2 may have arbitrary policies in different episodes.

Theorem K.1 mainly follows from the following Lemma K.6 and K.7. Select $m = \frac{512 S H^2 (\log \log H)^2 \log_2\left(8T^2 S H\right) \ln(6/\delta_1)}{\varepsilon^2}$, $\delta_1 := \frac{\delta}{2U_{max}S}$, $U_{max} := SA \log_2 T$ and $w_{\min} := \frac{\varepsilon}{4HSA}$ for any $0 < \varepsilon < H$, and any $0 < \delta < 1$ and then we have the following two lemmas.

**Lemma K.6.** *(cf. Lemma 2 in [9]) Let $E$ be the number of episodes $i$ for which there are $\kappa$ and $\iota$ with $|X_{i,\kappa,\iota}| > \kappa$, i.e. $E = \sum_{i=1}^{\infty} \mathbb{1}\{\exists(\kappa, \iota) : |X_{i,\kappa,\iota}| > \kappa\}$ and assume that $m \geq \frac{6H^2}{\varepsilon} \ln(2E_{\max}/\delta)$, where $E_{\max} = \log_2\left(\frac{H}{w_{\min}}\right) \log_2(SA)$. Then $\mathbb{P}(E \leq 6SAE_{\max}m) \geq 1 - \delta/2$.*

*Proof.* The proof mainly follows as Lemma 2 [9]. Here we point out the differences between the original UCFH algorithm [9] and our UCSG, when we remove the input $\varepsilon$.

1. Their stopping rule for phase $k$ is dependent on the specification of $\varepsilon$.

2. They set an upper bound for the maximum number of executions for each state-action pair $(s, a)$, which is determined beforehand and hardcoded in their algorithm.

3. Our algorithm only needs input $\delta$ to specify the failure probability and has $(\varepsilon, \delta)$-PAC bounds for arbitrarily selected $\varepsilon$.

The original UCFH nearly doesn't need the parameter $\varepsilon$ except at one place: their phases stops when "$\exists (s, a), v_k(s, a) \geq \max\{mw_{\min}, n_k(s, a)\}$ and $n_k(s, a) < SmH$." Since $w_{\min}$ and $m$ are defined through $\varepsilon$, this stopping rule requires $\varepsilon$ to be known by the algorithm. They need this because they would like to control $U_{\max}$, the total number of phases run by the algorithm. In their case, having this stopping rule, $U_{\max} \leq SA \log_2 \frac{SmH}{mw_{\min}} = SA \log_2 \frac{SH}{w_{\min}}$ because phase change won't be triggered when $n_k(s, a) < mw_{\min}$ or $n_k(s, a) > SmH$. However, since we assume that the time horizon $T$ is known, we can simply use $U_{\max} \leq SA \log_2 T$, and this can simplify our stopping rule to only "$\exists (s, a), v_k(s, a) \geq n_k(s, a)$."

Therefore, we can totally abandon the use of $\varepsilon$ in our algorithm, but enjoy their analysis results. The results automatically hold for arbitrarily selected $\varepsilon$. However, since we bound the number of $\kappa$ by $\log_2(4HSAT/\varepsilon)$ in Lemma K.7, we cannot let $\varepsilon$ tends to 0 too fast. (The minimum $\varepsilon$ we will select is $\varepsilon_0 = \min\{H, \sqrt{(H^3 S^2 A)/T}\}$ as in the proof of Lemma 6.3, where we select $H = \max\{D, \sqrt[3]{D^2 T/(S^2 A)}\}$ for Theorem 3.2 ).

$\square$

**Lemma K.7.** *(cf. Lemma 3 in [9]) Assume $M \in \mathcal{M}_k$. If $|X_{i,\kappa,\iota}| \leq \kappa$ for all $(\kappa, \iota)$ and for all $0 < \varepsilon \leq 1$ and $m \geq 512 \frac{CH^2}{\varepsilon^2}(\log_2 \log_2 H)^2 \log_2\left(\frac{4HSAT}{\varepsilon}\right) \log_2(SA) \ln(6/\delta_1)$. Then $|V_H(M_k^1, \pi_k^1, \pi_i^2) - V_H(M, \pi_k^1, \pi_i^2)| \leq \varepsilon$.*

*Proof.* It mainly follows the same proof as Lemma 3 in [9]. It was shown sufficient to let $m \geq 512 C(\log_2 \log_2 H)^2 |\mathcal{K} \times \mathcal{I}| \frac{H^2}{\varepsilon^2} \ln(6/\delta_1)$. The only differences are in the upper bounds for $|\mathcal{K} \times \mathcal{I}|$. In UCFH, the maximum number of executions of each state-action pair is set equal to $mSH$. Thus their knownness $\kappa(s, a)$ is no more than $\frac{n(s,a)}{mw_{min}} \leq \frac{4S^2 AH^2}{\varepsilon}$, whereas in our setting, since $n(s, a) \leq T$, $\kappa(s, a) \leq \frac{n(s,a)}{mw_{min}} \leq \frac{4HSAT}{\varepsilon}$. Thus in our setting $|\mathcal{K} \times \mathcal{I}| \leq \log_2\left(\frac{4HSAT}{\varepsilon}\right) \log_2(SA)$. $\square$

*Proof of Lemma 6.3.* Let $\varepsilon_0 := \min\left\{H, \sqrt{(H^3 S^2 A)/T}\right\}$, and $\delta_0 := \delta/\lceil \log_2(H/\varepsilon_0) \rceil$. We invoke logarithmically many times the bound in Theorem K.1 and use the union bound to obtain the regret. By assumption, for $j = 1, ..., \lceil \log_2(H/\varepsilon_0) \rceil$, with probability no less than $1 - \delta_0$, there are at most $\tilde{\mathcal{O}}(4^j S^2 A)$ episodes that are not $(2^{-j} H)$-optimal. Then the total error is bounded by

$$\sum_k \sum_{i \in \text{ph}(k)} \left| V_H(M_k^1, \pi_k^1, \pi_i^2, s_{\tau_i}) - V_H(M, \pi_k^1, \pi_i^2, s_{\tau_i}) \right|$$

$$:= \sum_k \sum_{i \in \text{ph}(k)} r_i = \sum_{i: r_i \leq \varepsilon_0} r_i + \sum_{i: r_i > \varepsilon_0} r_i$$

$$\leq \varepsilon_0 \frac{T}{H} + \sum_{j=1}^{\lceil \log_2(H/\varepsilon_0) \rceil} \tilde{\mathcal{O}}(4^j S^2 A)(2^{-j+1} H)$$

$$= \varepsilon_0 \frac{T}{H} + 4(2^{\lceil \log_2(H/\varepsilon_0) \rceil} - 1)\tilde{\mathcal{O}}(HS^2 A)$$

$$\leq \varepsilon_0 \frac{T}{H} + 8\frac{\tilde{\mathcal{O}}(H^2 S^2 A)}{\varepsilon_0} = \tilde{\mathcal{O}}(S\sqrt{HAT} + HS^2 A). \quad \square$$

## L  Proofs for Offline Training Complexity

*Proof of Theorem 7.1.* Define

$$K_\varepsilon := \{k : \rho^*(M) - \min_{\pi^2} \rho(M, \pi_k^1, \pi^2) > \varepsilon\},$$

$$K'_\varepsilon := K_\varepsilon \cap \{k : \text{phase } k \text{ is benign}\}.$$

Also, define

$$\text{Reg}_\varepsilon^{(\text{off})'} := \sum_{k \in K'_\varepsilon} T_k(\rho^*(M) - \min_{\pi^2} \rho(M, \pi_k^1, \pi^2))$$

$$= \text{Reg}_\varepsilon^{(\text{on})'} + \sum_{k \in K'_\varepsilon} \sum_{t=t_k}^{t_{k+1}-1} \left( r_t - \min_{\pi^2} \rho(M, \pi_k^1, \pi^2) \right).$$

(46)

where $\text{Reg}_\varepsilon^{(\text{on})'}$ is defined as a summation similar to $\text{Reg}_T^{(\text{on})}$ except that it is summed only over time steps in phases $k \in K'_\varepsilon$. Besides, analogous to the definition of $L_\varepsilon$, we define $L'_\varepsilon := \sum_{k:\text{benign}} T_k \mathbb{1}\{\rho^*(M) - \min_{\pi^2} \rho(M, \pi_k^1, \pi^2) > \varepsilon\}$.

We will argue (a) the order of $\text{Reg}_\varepsilon^{(\text{off})'}$ does not exceed that of $\text{Reg}_\varepsilon^{(\text{on})'}$, and (b) the upper bound of $\text{Reg}_\varepsilon^{(\text{on})'}$ is similar to that of $\text{Reg}_T^{(\text{on})}$ except that the dependency on $T$ is replaced by $L'_\varepsilon$.

To show (a), we note that the extra terms in $\text{Reg}_\varepsilon^{(\text{off})'}$ compared to $\text{Reg}_\varepsilon^{(\text{on})'}$ are the sum of

$$\sum_{t=t_k}^{t_{k+1}-1} \left( r_t - \min_{\pi^2} \rho(M, \pi_k^1, \pi^2) \right) = \sum_{n=5}^{7} \Lambda_k^{(n)},$$

$$\Lambda_k^{(5)} := \sum_{t=t_k}^{t_{k+1}-1} \left( r_t - \rho(M, \pi_k^1, \pi_k^2) \right),$$

$$\Lambda_k^{(6)} := \sum_{t=t_k}^{t_{k+1}-1} \left( \rho(M, \pi_k^1, \pi_k^2) - \rho(M_k^2, \pi_k^1, \pi_k^2, s_{t_k}) \right),$$

$$\Lambda_k^{(7)} := \sum_{t=t_k}^{t_{k+1}-1} \left( \rho(M_k^2, \pi_k^1, \pi_k^2, s_{t_k}) - \min_{\pi^2} \rho(M, \pi_k^1, \pi^2) \right),$$

over $k \in K'_\varepsilon$. $\Lambda_k^{(7)}$ is bounded by $T_k \gamma_k$ by (6); the bound of this term is the same as that of $\Lambda_k^{(1)}$. $\Lambda_k^{(5)}$ and $\Lambda_k^{(6)}$ are symmetric to $\Lambda_k^{(4)}$ and $\Lambda_k^{(3)}$ respectively (note that the $\bar\pi_k^2$ we constructed in Section 5.1 will be identical to $\pi_k^2$ in the offline setting). Therefore, we can use the same bounds for the corresponding terms.

Now we proceed to argue (b) and bound $\text{Reg}_\varepsilon^{(\text{on})'}$. We will largely reuse the regret analysis we already done for $\text{Reg}_T^{(\text{on})}$, but only sum up the contribution from phases in $K'_\varepsilon$.

The contribution to $\text{Reg}_\varepsilon^{(\text{on})'}$ from $\Lambda_k^{(1)}$ is

$$\sum_{k \in K'_\varepsilon} T_k \gamma_k = \sum_{k \in K'_\varepsilon} T_k / \sqrt{t_k};$$

(47)

the contribution from $\Lambda_k^{(3)}$ is as shown in (42):

$$\sum_{k \in K'_\varepsilon} \sum_{s,a} \frac{v_k(s,a)}{\sqrt{n_k(s,a)}} \tilde{\mathcal{O}}(D\sqrt{S}) + \sum_{k \in K'_\varepsilon} \tilde{\mathcal{O}}(DS);$$

(48)

finally, the contribution from $\Lambda_k^{(4)}$ is as shown in (34):

$$\sum_{k \in K'_\varepsilon} \left( \bar{h}_k(s_{t_{k+1}-1}) - \bar{h}(s_{t_k}) + \sum_{t=t_k}^{t_{k+1}-1} (Y_t^1 + Y_t^2) \right).$$

(49)

$\text{Reg}_\varepsilon^{(\text{on})'}$ is then bounded by the sum of (47)-(49). By lemma I.1, (47) is bounded by $(\sqrt{2}+1)\sqrt{L'_\varepsilon}$, and the first term in (48) is bounded by $\tilde{\mathcal{O}}(\sqrt{SAL'_\varepsilon})\tilde{\mathcal{O}}(D\sqrt{S}) = \tilde{\mathcal{O}}(DS\sqrt{AL'_\varepsilon})$ by Cauchy inequality.

The second term in (48) can be still bounded by $\tilde{\mathcal{O}}(DS^2A)$. Since the martingale difference sequences in (49) are now summing over a total of $L'_\varepsilon$ steps, (49) is now bounded by $DSA + D\sqrt{SL'_\varepsilon}$ (cf. (36)).

As a whole, we conclude that $\mathrm{Reg}_\varepsilon^{(\mathrm{on})\prime} \leq \tilde{\mathcal{O}}(DS\sqrt{AL'_\varepsilon} + DS^2A)$, and hence $\mathrm{Reg}_\varepsilon^{(\mathrm{off})\prime} \leq \tilde{\mathcal{O}}(DS\sqrt{AL'_\varepsilon} + DS^2A)$ by the argument in (a).

Note that by the definition of $K'_\varepsilon$, we have

$$
\begin{aligned}
\mathrm{Reg}_\varepsilon^{(\mathrm{off})\prime} &= \sum_{k \in K'_\varepsilon} T_k(\rho^*(M) - \min_{\pi^2} \rho(M, \pi_k^1, \pi^2)) \\
&\geq \sum_{k \in K'_\varepsilon} T_k\varepsilon = \varepsilon L'_\varepsilon.
\end{aligned}
\tag{50}
$$

Combining (50) with the upper bound of $\mathrm{Reg}_\varepsilon^{(\mathrm{off})\prime}$ just established, we have

$$
\varepsilon L'_\varepsilon \leq \tilde{\mathcal{O}}(DS\sqrt{AL'_\varepsilon} + DS^2A),
$$

which has the solution

$$
L'_\varepsilon \leq \tilde{\mathcal{O}}\left(\frac{D^2S^2A}{\varepsilon^2}\right).
$$

Comparing the definitions of $L_\varepsilon$ and $L'_\varepsilon$, and by Lemma 5.3, we get

$$
L_\varepsilon \leq L'_\varepsilon + \tilde{\mathcal{O}}(D^3S^5A) = \tilde{\mathcal{O}}\left(D^3S^5A + \frac{D^2S^2A}{\varepsilon^2}\right).
$$

Finally, we remark on how to select a single stationary policy after we have run the algorithm for $T$ steps. Note that in our proofs, we actually bound the single step regret in phase $k$ through

$$
\rho^*(M) - \min_{\pi^2} \rho(M, \pi_k^1, \pi^2) \leq \min_{\pi^2} \rho(M_k^1, \pi_k^1, s_{t_k}) - \rho(M_k^2, \pi_k^1, \pi_k^2, s_{t_k}) + 2\gamma_k
\tag{51}
$$

because LHS is $\frac{1}{T_k}\sum_{n=1}^{7} \Lambda_k^{(n)}$ while RHS is $\frac{1}{T_k}\sum_{n=2}^{6} \Lambda_k^{(n)} + 2\gamma_k$. Note that the terms on RHS can all be obtained by the algorithm, so they form an available upper bound for the LHS. Let $u_k$ denotes the RHS. Then the previous proofs actually proved that

$$
\sum_k T_k \mathbb{1}\{u_k > \varepsilon\} \leq \tilde{\mathcal{O}}\left(D^3S^5A + \frac{D^2S^2A}{\varepsilon^2}\right)
$$

holds with high probability. Therefore, if $T > \tilde{\Omega}\left(D^3S^5A + \frac{D^2S^2A}{\varepsilon^2}\right)$, there will be some $k$ such that $u_k < \varepsilon$. Since the algorithm knows $u_k$, it can just select the minimum of all $u_k$'s among all phases. That will output a policy $\pi_k^1$ such that $\rho^*(M) - \min_{\pi^2} \rho(M, \pi_k^1, \pi^2) \leq \varepsilon$.

$\square$

*Proof of Theorem 7.2.*

$$
\begin{aligned}
\mathrm{Reg}_\varepsilon^{(\mathrm{off})} &:= \sum_{k \in K_\varepsilon} T_k(\rho^*(M) - \min_{\pi^2} \rho(M, \pi_k^1, \pi^2)) \\
&= \mathrm{Reg}_\varepsilon^{(\mathrm{on})} + \sum_{k \in K_\varepsilon} \sum_{t=t_k}^{t_{k+1}-1} \left(r_t - \min_{\pi^2} \rho(M, \pi_k^1, \pi^2)\right),
\end{aligned}
$$

where $\mathrm{Reg}_\varepsilon^{(\mathrm{on})}$ is the sum of $\Delta_k$ over $k \in K_\varepsilon$. Of the six regret terms (5), $\Delta_k^{(4)}$ dominates over $\Delta_k^{(1)}$, $\Delta_k^{(3)}$, $\Delta_k^{(5)}$, and $\Delta_k^{(6)}$. So we only look at the $\Delta_k^{(2)}$ and $\Delta_k^{(4)}$. $\Delta_k^{(2)}$ is bounded by $\frac{T_k}{H}D + T_k\gamma_k$. Summing over $k \in K_\varepsilon$ by Lemma I.1 gives $\frac{L_\varepsilon}{H}D + \tilde{\mathcal{O}}(\sqrt{L_\varepsilon})$. Thus its average error is bounded by $\tilde{\mathcal{O}}(D/H + 1/\sqrt{L_\varepsilon})$. By taking $H = D/(2\varepsilon)$ we have the sample complexity for the second term is $\tilde{\mathcal{O}}(1/\varepsilon^2)$. On the other hand, by Theorem K.1, $\Delta_k^{(4)}$ has sample complexity bound $\tilde{\mathcal{O}}(HS^2A/\varepsilon^2)$.

By substituting $H = D/(2\varepsilon)$ gives the dominating sample complexity bound $\tilde{\mathcal{O}}(DS^2A/\varepsilon^3)$. We argue again the order of $\text{Reg}_\varepsilon^{(\text{off})}$ does not exceed that of $\text{Reg}_\varepsilon^{(\text{on})}$. To show this, we note that the extra terms in $\text{Reg}_\varepsilon^{(\text{off})}$ compared to $\text{Reg}_\varepsilon^{(\text{on})}$ are the sum of

$$\sum_{t=t_k}^{t_{k+1}-1} \left( r_t - \min_{\pi^2} \rho(M, \pi_k^1, \pi^2) \right) = \sum_{n=7}^{11} \Delta_k^{(n)},$$

$$\Delta_k^{(7)} := \sum_{i \in \text{ph}(k)} \left( \sum_{t=\tau_i}^{\tau_{i+1}-1} r(s_t, a_t) - V_H(M, \pi_k, s_{\tau_i}) \right),$$

$$\Delta_k^{(8)} := \sum_{i \in \text{ph}(k)} \left( V_H(M, \pi_k, s_{\tau_i}) - V_H(M_k^2, \pi_k, s_{\tau_i}) \right),$$

$$\Delta_k^{(9)} := \sum_{i \in \text{ph}(k)} \left( V_H(M_k^2, \pi_k, s_{\tau_i}) - H\rho(M_k^2, \pi_k, s_{\tau_i}) \right),$$

$$\Delta_k^{(10)} := \sum_{i \in \text{ph}(k)} \left( H\rho(M_k^2, \pi_k, s_{\tau_i}) - H\min_{\pi^2} \rho(M, \pi_k^1, \pi^2, s_{\tau_i}) \right),$$

$$\Delta_k^{(11)} := 2H,$$

over $k \in K_\varepsilon$. This decomposition mirrors that in (5) where $\Delta_k^{(7)}, \Delta_k^{(8)}, \Delta_k^{(9)}, \Delta_k^{(10)}$ and $\Delta_k^{(11)}$ are symmetric to the $\Delta_k^{(5)}, \Delta_k^{(4)}, \Delta_k^{(2)}, \Delta_k^{(1)}$, and $\Delta_k^{(6)}$ in (5), respectively, and we can use the same bounds for the corresponding terms.

Finally, we can pick an $\varepsilon$-optimal policy $\pi_k^1$ after the algorithm has run for $T > \tilde{\mathcal{O}}\left( \frac{DS^2A}{\varepsilon^3} \right)$ steps. The way is similar to that described in the proof of Theorem 7.1. $\qquad \square$

## M  Other Technical Lemmas

**Remark M.1.** *Under Assumption 1, note that for any stationary policy $\pi$, we have $\text{sp}(h(M, \pi, \cdot)) \leq T^\pi(M)$. Indeed,*

$$h(M, \pi, s) = \mathbb{E}_s^\pi \left[ \sum_{t=1}^\infty r_t - \rho(M, \pi) \right]$$

$$\leq T_{s \to s'}^\pi(M) + \mathbb{E}_{s'}^\pi \left[ \sum_{t=1}^\infty r_t - \rho(M, \pi) \right]$$

$$= T_{s \to s'}^\pi(M) + h(M, \pi, s').$$

**Remark M.2.** *Imagine an MDP where all transitions from $s \neq s'$ remain the same while $s'$ becomes an absorbing state; rewards on $s \neq s'$ are all 1 and 0 on $s'$. Now $\max_{\pi^1} \max_{\pi^2} T_{s \to s'}^{\pi^1, \pi^2}(M)$ is equivalent to the maximum reward on this MDP, which can be achieved by stationary joint policy by both players.*

## N  Regularization/Constraint-based Approach for Assumption 1

It is possible to improve the $\tilde{\mathcal{O}}(D^3S^5A)$ term in the regret bound under Assumption 1. Note that this term mainly comes from Lemma H.3, which says that to wait until $\text{sp}(h(M_k^1, \pi_k^1, \bar{\pi}_k^2, \cdot)) < 2D$, we need to pay $\tilde{\mathcal{O}}(D^3S^5A)$ regret. However, if we can know the value of $D$ in advance, the optimistic model $M_k^1$ can be selected based on the following constrained optimization problem:

$$M_k^1 = \underset{\tilde{M} \in \mathcal{M}_k}{\arg\max} \max_{\pi^1} \min_{\pi^2} \rho(\tilde{M}, \pi^1, \pi^2, s_{t_k}),$$

$$\text{subject to} \quad \forall \pi^1, \pi^2 \in \Pi^{\text{SR}}, \text{sp}(h(\tilde{M}, \pi^1, \pi^2, \cdot)) \leq D.$$

Clearly, the true model $M$ still lies in this feasible set, so this is a valid way to select $M_k^1$. It is also possible to convert this into a regularized optimization problem as demonstrated by [3]. Nevertheless,

we are not aware of any practical algorithm that can solve either optimization problem. We just demonstrated in this paper that the benefit of this regularization/constraint-based approach is only on the additive constant but not on the asymptotic performance.