[Reviews · NeurIPS 2017]

Reviewer 1



The paper considers the problem of online learning in two-player zero-sum stochastic games. The main result is constructing a strategy for player 1 that guarantees that the cumulative rewards will never go below the maximin value of the game by more than a certain bound, no matter what strategy the other player follows. The bound is shown to grow sublinearly in the number of rounds T of the game, and polynomially on other problem parameters such as the diameter, the size of the state and action spaces. The results imply that the proposed algorithm can be used in self-play to compute near-maximin strategies for both players. The algorithm and the analysis are largely based on the UCRL algorithm of Auer and Ortner (2007) and the analysis thereof. I have reviewed a previous version of the paper for ICML'17, and have to say that the quality of the writing has improved quite a bit; the authors obviously invested a lot of effort into improving the readability of the paper. The paper is of course still very technical with most of the analysis details deferred to the appendix. Being already familiar with the analysis, I cannot objectively judge how easy it is to follow for first-time readers (I believe that it still looks rather intimidating), but I am now entirely convinced that the proofs are correct. Since the previous round of reviews, the authors fixed all the technical issues I've pointed out, so at this point, I have no technical objections. The contribution itself is novel and interesting: the paper proposes a new (but very natural) regret definition suitable for stochastic games and proposes an efficient algorithm that comes with strong guarantees on the proposed regret notion. The fact that such strong results are actually possible was quite surprising to me. All in all, I think that this paper is ready to be published. Detailed comments ================= -definition of \rho^* after line 83: at this point, the existence of the limit is not ensured. Please at least say that this requires some mild assumptions that will be specified later. -112: Assumptions 1 and 2: Maybe it would be useful to use different letters for the two diameter notions to avoid confusion. -154: "Player 1 optimistically picks one model"---Well, not entirely optimistically, as the choice of the opponent's policy is done conservatively. Please rephrase for clarity. -186: This Lemma 4 is very informal; please state explicitly that this is not the full statement of the lemma. Also, the statement itself is written a bit strangely. -189: "analyzing techniques" -> "analysis techniques" -195: "our algorithm makes [...]" -> "our algorithm uses policy [...]" -261: Please highlight the fact that \pi_i^2 is a non-stationary policy, and in general all policies discussed in this section can be non-stationary.

Reviewer 2



This paper proposes a learning algorithm for two-player zero-sum games with a better offline sample complexity bound than previous algorithms, in the average reward setting. The analysis of bounds is based  on an innovative way to replace the opponent's non-stationary policy with a stationary policy. The paper looks good and seems to present a significant progress over the state of the art. It is written extremely technically and concisely. However, there is no clear separation between the main paper and the supplementary material (the former depends heavily on the latter) and the whole paper is 33 pages long. The content is too technical for me and I cannot evaluate its validity, but I'm quite convinced that NIPS is not an appropriate target for this work, which should be submitted to a journal instead. My remarks below are mainly clarification requests from a naive reader who did his best to understand some of the content (and could not). line 79, what does R stands for in \PI^{HR} and \PI^{SR}? line 82: it is standard to have M a a parameter of \rho? You may suggest that later you will use some approximation \hat{M} here... line 85: I could not get what are the coordinates of vector h. "Coordinatewisely" is a too compact neologism ;) Note that is other domains, vectors are written in bold, this helps decoding equations... line 87: the Bellman equation => Bellman's equation (?) Getting the given equation from the standard one is probably trivial, but you should either explain it or give a reference on how to get it. Notations: this part is a nightmare. You didn't define the notion of game matrix before. I don't know what "the probability simplex of dimension k" is. I could not get any clear message from this small section. BTW, shouldn't it come earlier? General comment: Lemmas, Definitions, Theorems, Remarks should each have their own counter. line 114: are as follows => is line 146: you should refer to Algorithm 1 somewhere in the text line 152: I don't understand "it can then create a confidence region Pk(s; a) for each transition probability". line 156: why do you need to double one count at each iteration? That is, why not triple or anything else? line 160: LHS: notation not introduced line 190: difficulties that (are) only present line 192: you already introduced H_t line 76... line 255: doesn't => does not line 259: .Now => . Now line 272: won't => will not After section 4, I find the paper unreadable, because I always have to go to the supplementary part. This is not a good organization. The main part of the paper stops abruptly without a discussion, a conclusion summarizing the main points and their significance, nor suggestions for future work.

Reviewer 3



The authors propose a reinforcement learning algorithm for stochastic games in the infinite-horizon average-reward case. The authors derive regret bounds for the algorithm both in the online and offline settings. The paper is easy to read and technically sound, although I did not check all the proofs. It seems that the regret bound of sqrt(T) is achieved only under the strong Assumption 1, while the more natural Assumption 2 only gives sqrt(T^{2/3}). It would be helpful if the authors could elaborate on whether sqrt(T) is achievable under Assumption 2, perhaps with a lower bound. ============================================================= Update after author feedback period ============================================================= The authors should compare the results with that of https://doi.org/10.1287/moor.2016.0779 In particular, the stochastic game setting can be related to the robust MDP setting with a finite number of uncertainty sets in the above paper, where a sqrt(T) regret bound is achieved under an assumption that I believe is (roughtly) between Assumption 1 and Assumption 2 in the present paper.